# Circular Bioeconomy Business Models to Overcome the Valley of Death. A Systematic Statistical Analysis of Studies and Projects in Emerging Bio-Based Technologies and Trends Linked to the SME Instrument Support

**Fabiana Gatto \* and Ilaria Re**

Consorzio Italbiotec, 20138 Milan, Italy; ilaria.re@italbiotec.it
\* Correspondence: fabiana.gatto@italbiotec.it

**Abstract:** Reducing the environmental pressure along the products life cycle, increasing efficiency in the consumption of resources and use of renewable raw materials, and shifting the economic system toward a circular and a climate-neutral model represent the heart of the current macro-trends of the European Union (EU) policy agendas. The circular economy and bioeconomy concepts introduced in the EU's Circular Economy Action Plan and the Bioeconomy Strategy support innovation in rethinking economic systems focusing on market uptaking of greener solutions based on less-intensive resource consumption. In recent decades, industrial research has devoted enormous investments to demonstrate sustainable circular bio-based business models capable of overcoming the "*Valley of Death*" through alternative strategic orientations of "*technological-push*" and "*market-pull*". The study highlights industrial research's evolution on bio-based circular business model validation, trends, and topics with particular attention to the empowering capacity of start-ups and small and medium-sized enterprises (SMEs) to close the loops in renewable biological use and reduce dependence on fossil fuels. The research methodology involves a bibliographic search based on the Preferred Reporting Items for Systematic Reviews and Meta-Analyses (PRISMA) approach and the European Innovation Council (EIC) Accelerator Data Hub investigation to understand SMEs' key success factors and start-ups of the circular bioeconomy sector. Eco and bio-based materials, nutraceuticals, and microalgae represent the most sustainable industry applications, leading to circular bioeconomy business models' future perspective.

**Keywords:** circular bioeconomy; circular bioeconomy business model; valley of death; SME; startup

## 1. Introduction

The transition from a linear economy to a circular one rapidly stimulates new business models, gaining in all production sectors, including the bio-based world.

Despite the bioeconomy contribution to tackling global climate challenges being widely recognized, many products and technologies with high potential do not reach the market, unsuccessfully overcoming the so-called "*Valley of Death*".

The term is conventionally used in the Venture Capitals environment and refers to the company's start-up phase, which is represented as an evolutionary curve of its financial performance ranging from initial capital availability to the break-even point's achievement of the production of profits.

The financing of a start-up embraces the high-risk phases of pre-seed and seed in which the capital comes from the company's founders, and the product has been designed or only prototyped. Incubators, business angels, grants, and subsidized loans within 1 million euros are the initiative's leading sources. Therefore, the *Valley of Death* coincides

with the demonstration phase of the model's feasibility and profitability, during which a start-up struggles to identify incremental risk capital toward the industrial up-scaling. Part of the business model's failure is related to the difficulty of reassuring investors about the market's acceptability or technology and the payback time credit.

According to literature studies, the *Valley of Death* curve's duration and characteristics vary according to several factors, such as the availability of initial liquidity, the solidity of the business plan, the organizational resources, the type of product and ability to demonstrate the market transfer potential, and the capacity to affect business angels and external experts in turning ideas into commercial innovations [1,2]. The strategic role of accelerators, business angels, and grants is relevant for bridging the resource gap and addressing the "*Fuzzy Front End of Innovation*" in pre-seed and seed phases, while Venture Capitals, equity, and debt capital support the transition from innovation to commercialization, helping to reduce the *Valley of Death* duration [2–4]. Established companies tackle this challenging phase more quickly and effectively than start-ups or SMEs thanks to long-standing market positioning and availability of financing and research infrastructures [5,6].

According to some studies, start-ups deeply depend on building trust relationships with business angels and other financial resource providers—including grants—to promote and place on the market new technologies and bioproducts [2,7]. Trust is a mechanism of relational risk regulation enabling business angels to tackle a particular form of agency risk described by Maxwell and related to the business angels' subjective evaluation of how entrepreneurs probably decide to spend their money [8]. According to Maxwell and Lévesque, four main factors define the development of business angels' trust in the investment phase: reliability, ability, behaviour, and communication [8,9].

In the last few decades, economic models with a high environmental and social impact capable of overcoming the *Valley of Death* have become central to policy tools and public funding development. The rapid development of the circular bioeconomy has emphasized the high business potential based on the enhancement of renewable biological resources; however, the crucial success factors remain poorly known such as the balance between supply-side and demand-side and the role played by the strategic orientations of "technology-push" to "market-pull" of new products and technologies.

In the field of circular bioeconomy, although more and more consumers are interested in recognizing a high value for sustainability, many SMEs and start-ups adopt non-innovative and solid business models, risking the *Valley of Death* followed by the inability to reach the market.

This study investigates the research transition toward testing and validation eco-industry circular business models successfully reaching the market to define outlooks and the most promising sector scenarios and trends. The work's novelty lies in an integrated approach that combines analysis of the literature and funded projects in the demonstration phase to identify the success factors of circular bio-based business models that could reach the market by overcoming the high-risk pre-seed and seed phases in the *Valley of Death*. Although based on disruptive innovation, many business models falling into these high-risk phases cannot overcome this gap, and then it is relevant to understand and communicate what might be the factors that limit the development of an innovative and sustainable idea. Raising the inventor's awareness and encouraging a trust relationship with the potential financial investor or business angel represent crucial tools.

The study's introduction offers an overview of the European legislation's pillars from the Strategy for the Bioeconomy and the Circular Economic Action Plan to stimulate new production paradigms based on a circular bioeconomy and financial instruments toward a real carbon-neutral economy. Challenges and opportunities related to the new concept of circular bioeconomy and the success factors in overcoming the *Valley of Death* are finally treated in the second part of the study. The evaluation of the "technology-push" and "market-pull" strategic orientations of the new business models is explored to define the

most relevant successful invoices for the commercial exploitation of disruptive innovations.

For this study's purpose, the methodology provides a first systematic analysis of the publications, through the PRISMA method and the European Innovation Council (EIC) Accelerator Data Hub investigation to identify the main successful circular business models. The results reported and analyzed in this study provide a qualitative and quantitative analysis of the obtained data. The selection of projects collected by the EIC Accelerator Data Hub is the starting point for the further classification of business models according to their "*technology-push*" and "*market-pull*" strategic orientations. Based on this first classification, the survey allows us to deepen and fully understand the business models' mechanism with a greater probability of success.

The discussion and subsequent conclusions aim to return the state of the art on the main circular bio-based business models and the possible application sectors and trends of the bioeconomy sector. In detail, the discussion returns a ranking of the main application sectors identified through the detailed project's analysis. Therefore, it allows presenting a first trend of expanding sectors that need projects with circular bio-based business models to reach the market safely and innovatively. Therefore, in conclusion, a real ranking is reported, based on both literature and business models, promising to identify the path of development of the circular bioeconomy.

## 1.1. From the Bioeconomy Strategy to the European Green Deal: The Policy Pathway toward a Greener European Economy

Increasing climate change, environmental degradation, and the consequent biodiversity loss have prompted Europe to shift the production system from a fossil-based and linear economy to a bio-based circular economy paradigm. In the last decades, European policy agendas and R&I (Research and Innovation) programs converged with bioeconomy and circular economy production and consumption models to reconcile environmental, economic, and socio-economic goals for climate-neutral and sustainable growth.

The bioeconomy or bio-based economy is the production of renewable biological resources and the conversion of these resources and waste streams into value-added products, such as food, feed, bio-based products, and bioenergy [10]. The transition to a bio-based economy is necessary to ensure that future generations have adequate resources and living conditions—so, in other words, to ensure sustainable growth. The bioeconomy, combining industrial production efficiency and the reduction of by-products and wastewater (no waste economy), makes an essential contribution to achieving this goal [11].

In 2012, the European Union launched the first bioeconomy strategy, which was updated in 2018, providing the opportunity to create a coherent political framework to accelerate the deployment of a sustainable European bioeconomy and maximize its contribution toward the 2030 Agenda and its Sustainable Development Goals (SDGs) [12]. The implementation of this strategy for the development of a sustainable and circular bioeconomy requires a joint effort from public authorities and industry, which have been called to support the new European Green Deal Investment Plan with 14 concrete measures launched in 2019 based on three key priorities:

- Strengthen and scale up the bio-based sectors, unlock investments and markets. Bioeconomy has the potential to innovate and modernize the European economy and industries. For this reason, it is essential to intensify the deployment of sustainable and circular biological solutions. The development of an investment platform dedicated to the circular bioeconomy with a financial contribution of 100 million euros will make it possible to bring bio-innovations closer to the market and facilitate the development of biorefineries and bioproducts.
- Deploy local bioeconomies rapidly across Europe. Facilitating the introduction of an EU support mechanism for bioeconomy policies will enable the Member States to

　　　　　establish national and regional programmes and launch pilot actions to develop bio-economies in rural, coastal, and urban areas.

- Understand the ecological boundaries of the bioeconomy. Climate change, pollution, soil degradation, and population growth are seriously undermining our ecosystem and represent a significant challenge. Implementing a system to monitor progress toward a circular and sustainable bioeconomy will help expand knowledge about specific bio-based processes and products. At the same time, it is crucial to promote the dissemination of good practices that will be used to guide the functioning of the bioeconomy within safe ecological limits.

After updating the Bioeconomy Strategy, some member countries (Portugal, France, Ireland, Italy, Germany, Austria, Finland, Latvia, United Kingdom) and EU regions have developed a roadmap for the bioeconomy and Smart Specialization Strategies. In this context, the idea of a global initiative focused on developing new value chains thanks to the bio-based industry has led to the BBI JU program (Bio-based Industries Joint Undertaking) with 3.7 billion euros of investments. Today, it represents the most remarkable example of industrial cooperation on a European level in the industrial biotechnology sector based on a public-private partnership model and destined to become the reference point of the bioeconomy [13]. For this reason, it is not surprising that industrial biotechnology is now entirely accepted as a "Key Enabling Technology" of the European Union or real engines for innovation to applied research. Industrial biotechnology, rightly considered the "key" for the development of the bioeconomy, can generate value from what is deemed to be worthless or even a cost to businesses (i.e., $CO_2$, biomass, or waste), transforming waste into a resource, according to the principles of the circular economy.

### 1.2. Bioeconomy and Circular Economy Symbiosis: Toward a New Sustainable, Productive Model

According to the Ellen Macarthur Foundation definition, the circular economy is "one that is restorative and regenerative by design and which aims to keep products, components and materials at their highest utility and value at all times, distinguishing between technical and biological cycles" [14]. In this context, the circular economy refers to a production system where the value of products, materials, and resources is maintained for as long as possible over time, protecting the environment, limiting emissions, and minimizing material losses.

The update of the Circular Economy Action Plan (March 2020), previously adopted in 2015 by the European Union with the Circular Economy Package, aimed to build cyclic and closed production systems. It also introduced measures to reduce the premature obsolescence of products, increase the percentage of recycled material, and encourage eco-design to facilitate the readjustment and renewal [15]. Finally, the reduction of the carbon footprint, single-use materials, and the increase in digitization are remarkable novelties of the plan.

Bioeconomy and circular economy go along with a value chain approach focusing on reducing the fossil raw materials dependence and $CO_2$ emissions, exploiting by-products. However, what are the main overlaps and differences between a bioeconomy and circular economy?

Bioeconomy and circular economy are two intersecting concepts with common overlaps, especially in sharing some global climate targets such as minimizing and accelerating fossil-based industries' conversion to low-carbon, resource-efficient, and sustainable ones [16]. The circular economy emphasizes the redesign of industrial processes to reducing inputs and outputs, keeping products' value up in the economy for as long as possible and increasing the eco-efficiency of processes. The bioeconomy tries to minimize fossil carbon mining, encouraging the use of renewable biological resources from agricultural, aquatic, and forestry sources and mitigating the climate change to find more sustainable bio-based alternatives. It addresses sustainable conversion processes such as biorefineries

and the cascading use of biomass, implicating a wide range of enabling industrial technologies and environmental impacts on food, feed, materials, and bioenergy production.

On the other hand, differences between bioeconomy and the circular economy are relevant in the same way. The majority of material flows such as biomass, metals, and minerals are not integrated into a cascading use of by-products; in fact, only 10–15% of the biomass in Europe is available to become part of this mechanism [16]. Therefore, a regenerative approach in production and consumption implied by a circular economy approach can not be fully adopted in some bio-based applications due to the impossibility of recycling or re-use energy, fuels, detergents, cosmetics, and coatings.

Achieving more recycling in biomaterials production is one of the main challenges of the bio-based sector. However, it requires further industrial effort to increase biodegradability and compostability proprieties through biomass cascading. Indeed, despite bio-based chemicals and bio-based fuels in common understandings being perceived as part of nature's carbon cycle, their conversion processes cannot avoid emissions. In Europe, 144 Mtoe of biomasses were consumed in 2017, which is equal to about 438 $MtCO_2$ saved in one year [17]. In this context, it talks about emissions avoided because biomass combustion is conventionally considered to be zero-emissions. Although combustion itself releases into the atmosphere the carbon contained in organic matter, these emissions shall be deemed to produce approximately the same amount of carbon dioxide as had been previously fixed by the same biomass through photosynthesis. In a greener economy scenario, the sustainable use of biomass for bio-based production, just as in the natural carbon cycle, must be carried out with adequate and innovative technologies to avoid or reduce other local pollutants emissions.

In fact, among the main drivers of bioeconomy and circular bioeconomy is reducing carbon emissions that lead to an increase in demand (and supply) for bio-based products. Bioeconomy, as a natural cyclical process, counteracts the natural resources degradation, loss of biodiversity, and ecosystem services, affecting the well-being of at least 3.2 billion people by costing the equivalent of about 10% of the world's annual gross product in 2010 [18]. Regarding processes, biorefineries, and advanced manufacturing involving biocatalysis and microorganisms (or parts of them, e.g., enzymes) have a pivotal role in achieving a more carbon-neutral economy [19].

Adopting a less resource-demanding economic system represents a critical pillar of the new European programme for the next decade. The Green New Deal, launched in December 2019, aims to make Europe the first climate-neutral continent by 2050 [20].

The Green New Deal foresees a ten-year program, in which the necessary financial instruments are provided to guarantee a correct ecological transition toward the strategic sectors of the circular bioeconomy and green economy. To achieve this objective, an investment plan of 1 trillion euros [21], between public and private resources, supports environmental-friendly technologies, industrial innovation, greener and cheaper mobility, energy sector decarbonization, the higher energy efficiency of buildings, and improved global environmental standards.

The commitment to climate neutrality achieving by 2050 becomes binding through a European law on climate, which also sets the milestone of the 55% reduction in emissions by 2030. The 2030 climate plan-integrated part of this roadmap is under negotiation between Parliament and the European Council and proposes financial measures and monitoring of the progress for the growing reduction of coal dependence and the socio-economic transformation of European coal regions. To support this revolution, the European Commission in 2019 proposed a Just Transition Fund (JTF) as the European Green Deal cornerstone with a budget of 43 billion euros.

Therefore, the ongoing EU budget negotiations for 2021–2027 tipped the balance to ensure the social inclusion and political acceptability of the EU decarbonization process.

All these proposals led to a new concept of the circular bioeconomy from the interaction between the bioeconomy and circular economy. According to a circular model ("no waste economy"), the sustainable use of biomass involves products that are processed, re-

used, and at the end of life reintegrated into the biosphere in the form of bio-based products. These two innovative and essential concepts, if joined together, could constitute a political, economic, and social instrument with a high value [16].

## 2. Start-Ups and SMEs as Key Actors into the Deployment of Successful Circular Bio-Based Business Models: A Hurdles Analysis in Facing the *Valley of Death*

The EU industrial innovation ecosystem is grounded in small and medium-sized enterprises (SMEs). They represent 99.8% of the industry, generating 4357 billion of value-added and 97.7 million jobs, respectively 56.4% and 66.6% of the total in 2018 [22]. A significant contribution came from specialized *knowledge-intensive services* and *high-tech manufacturing* sectors, driving 33% of SMEs value added in 2018. Micro firms accounted for 93% and were responsible for 18.3% of the increasing value added from 2014 to 2018. Bio-based is one of the most representative research-intensive sectors in Europe, in which SMEs, start-ups, and spin-offs, in particular, are growing.

The EU bio-based sector gained a turnover of 2.4 trillion euros in the EU-28 (European Union of 28 member states) in 2017 with a growth of 25% since 2008 [23]. Roughly half of the 2.4 trillion euros comes from the food and beverages sector; nearly 20% of the turnover is produced by the primary sectors (agriculture and forestry). Eco-industries are responsible for the 30% remaining, including chemicals and plastics, pharmaceuticals, paper and paper products, forest-based industries, the textile sector, biofuels and bioenergy, with 23% turnover increasing 600 million in 2008 to around 750 million euros in 2017 [23].

A booming trend in industrial research studies on circular bio-based models suggests a more significant effort of bioeconomy on shifting toward a circular approach. Small and medium-sized enterprises (SMEs) and start-ups play a pillar role in demonstrating their availability. Despite the breakthrough innovation potential, significant hurdles must be overcome to exploit greener products [24]. The success of bio-based products over fossil-based products is hampered by several factors, including high production costs, consumer awareness of the related benefits and low investor confidence in high-risk models. One of the most common barriers that start-ups face is the so-called "*Valley of Death*", which is the gap between the innovative research studies and profitable commercial exploitation. Hurdles in reaching the market and successfully attracting private investors are due to limited financial resources and organizational skills, together with risks associated with early-stage (unproven and proven technologies) and middle-stage (pre-commercial) technologies [25]. Critical market acceptability affects the last phases of the process, in which the scalability of the process and the Technology Readiness Level (TRL) have to be demonstrated [26]. Finally, an inherent problem linked to the local availability of biomass supply suitable for continuous processes, transformation cost, and transformation conditions represents the critical competitive factor for overcoming the "*Valley of Death*". It occurs when technology does not reach the market despite the high potential already tested on a laboratory scale.

Technological innovation is often indicated as the result of both "*technology-push*" and "*market-pull*" strategic orientation and in particular, the overlap and interaction between them. The two main guides for technological innovation on the right economic and institutional direction are represented by science and technology (push) and acknowledgement markets (pull) [27].

In other words, the "*market-pull*" is a situation in which the market demands a product (or service) type or defines a problem, so designers and producers make a product to meet that need. Therefore, the design and development of the market product are based on a specific request for which products or services the customer needs, which is intended to fill a market-defined niche. In opposition, "technology-push" is the state in which the producer creates a product type and the demand for that type. The technology push product development is mainly based on the belief that the supplier recognizes a market need even before the market does [28].

In the past decades, emerging studies have contributed to the enrichment of the circular bioeconomy literature. However, the industrial application of this concept requires innovative and solid business models [29]. Current scientific research focuses on technology-related research and bio-based technology research, but specific studies on the innovation of the circular business model related to overcoming the *Valley of Death* have not been found. Moreover, added to this is the evident scarcity of academic and practical approaches to circular business models and bioeconomy business models [30,31]. All this information has directed research toward the correlation of the circular business model concept and overcoming the *Valley of Death*.

In the bioeconomy sector, and even more in the circular bioeconomy, the trade-off between the "*market-pull*" and "*technology-push*" approach plays a crucial role in the success of new bio-based business models for start-ups or SMEs [4]. There are many innovative and excellent ideas, bioproducts, and green processes born and developed around bioeconomy megatrend. However, on the other hand, a reflection on all these new bio-based processes/products arises spontaneously: how many will manage to reach a large and consolidated market?

A tangible example of technology-push orientation is founded in bioplastics production. Today, there are several alternative bioplastics for all conventional plastics and their applications. Bioplastics—plastics that are bio-based and biodegradable—have the same properties as traditional plastics and offer additional benefits, such as resource depletion and environmental impact. Despite bioplastics currently representing about 1% of more than 359 million tons of plastic produced annually [32], the demand is rising, and the market for bioplastics is continuously growing and diversifying. It is currently worth $6.04 billion and is expected to reach $19.93 billion in 2026 [33]. This continuous diversification, frequently matching new technological applications, does not often follow market demand. An example is the PHA (polyhydroxyalkanoates), one bio-based and biodegradable biopolymer, representing 1.2% of the global bioplastic and less than 1% of the total bioplastic market value (57 million dollars) [32,34]. This biopolymer's limited commercialization is linked to several disadvantages hurdling its competitiveness with traditional synthetic plastics, including low chemical-physical properties, high production costs, and the difficulty of achieving high productivity.

Conversely, bio-based aromatics molecules' production—aromatic compounds derived from bio-based resources—meet the market-pull strategic orientation. Today, aromatics are essential building blocks for the chemical industry, and no less than 40% of chemicals are aromatic [35]. This developing sector promises to produce aromatics at a lower price than the oil industry, in a sustainable way, and it could shake up the chemical industry by maximizing the concept of market pull. These concrete examples support the hypothesis that it is crucial and almost essential to invest in a market to overcome the *Valley of death*.

Targeted EU measures aim to guarantee policy and financial support to increase SMEs' capability to turn cutting-edge ideas into business opportunities, bridging the gap between research and commercially viable products. Since the European Commission launched the Bioeconomy Strategy (2012) and the Circular Economy Package (2015), a progressive acceleration for a more sustainable, productive paradigm has been enforced. Under the Horizon 2020, the EU primary R&I EU Programme (2014–2020), a budget of over 500 million euros has been devoted to supporting the Industrial Leadership in the Biotechnology sector in which most topics contribute to the bioeconomy growth.

To encourage SMEs to fully implement their innovative business model without falling into the *Valley of Death*, the EU launched the EIC Accelerator Pilot or previously known SME Instrument. This European instrument functions as a catalyst for SMEs and start-ups supporting high-potential SMEs to develop ground-breaking innovative products, services, or processes ready to face global market competition [36]. The SME Instrument was designed (article 22 of Regulation (EU) n 1291/2013) to support SMEs during the different innovation cycle phases. Characterized by open calls, it is divided into multiple types of

support, such as exploring the feasibility and commercial potential of the project idea (Phase 1), testing, demonstration, and market replication activities (Phase 2), and enhancing commercial exploitation of the results, including access to private investors sources (Phase 3). It provided as a lump-sum grant for Phase 1 of 50,000 euros, covering 70% of the overall investment. The grant for Phase 2, usually between 0.5 and 2.5 million euros, covers up to 70% of eligible costs—100% if it has an intense research component. Phase 3 helps plan reinforcement to place one or several innovations (product, process, service) on the market. SME Instrument Phase II evolved in the EIC Accelerator Pilot in June 2019. Calls based on a more open bottom-up approach (previously limited to predefined subject areas) were introduced, as well as an improved selection process with face-to-face interviews with a group of experienced innovators. Finally, the program was also further improved, offering blended finance in the form of optional investments in equity and the grant of up to 15 million euros. Since the start of the program in 2014, the SME Instrument has helped over 4500 companies by providing up to €2.5 million in funding and tailored business innovation and acceleration services through 750 international coaches. More than two-thirds of the companies have placed their product on the market, and a total of 3 billion euros of extra private investment has been leveraged [37].

## 3. Materials and Methods

The research work is based on the Preferred Reporting Items for Systematic Reviews and Meta-Analyses method (PRISMA), a tool developed in 2005 by a group of clinical researchers to improve systematic reporting reviews and meta-analyses [38].

The PRISMA Statement consists of a 27-item checklist and a four-phase flow diagram, through which it was possible to conduct a funnel approach. In the present study, the PRISMA method supports the research of articles and publications focused on the circular bioeconomy concept, as the application of circularity or regenerative transformation of renewable biological resources in advanced bio-based products such as food, feed, bioenergy able to close to the loop of waste.

The circular bioeconomy application in manufacturing aims to create new business models valorizing biomass and bio-waste to realize high added-value products. Led by a primary challenge to reduce the demand for fossil carbon and valorize waste and side streams, the circular bioeconomy has the capacity to reach the UN Sustainable Development Goals. Despite this potential, several challenges on bio-based products development from laboratory to market have to be tackled. Indeed, disruptive technologies have to face critical issues in testing at large scale, regulations compliance, attracting public and private funds, and finally, the consumers' acceptability.

With the aims to analyze the more recent research trends in unlocking the potential of circular bioeconomy business models, including their connection with the Horizon 2020 investments and industrial applications, this study relied on specific publications from 2014 to 2020.

To understand how SMEs capture value through circular bioeconomy business models, especially in crossing the *Valley of Death* of the most promising solutions, the EIC Accelerator Data Hub is examined. It is an interactive tool developed by the Executive Agency for SMEs that generates information on SMEs' participation in EIC Accelerator (previous SME Instruments, a funding scheme part of the *Industrial Leadership* pillar of the Horizon 2020). This repository analyzes SMEs' support funding and identifies an innovative case study of the circular bioeconomy business model in Europe and Italy. In the context of the present study, it supported the qualitative analysis of successful circular bioeconomy business models.

### 3.1. Search and Selection Criteria for Systematic Literature Review

The systematic literature study uses two different databases: ScienceDirect and Scopus. ScienceDirect is the leading platform of peer-reviewed literature that hosts 16 million articles, 2500 journals, 370 open access journals, 39,000 books, and 330,000 topic pages [39].

Results are grouped into Physical Sciences and Engineering, Life Science, Health Sciences, Social Science, and Humanities. Scopus is the largest global and interdisciplinary database of abstracts and citations of peer-reviewed literature. It covers three sources: books, journals, and specialized magazines, ranging from natural sciences to social sciences, from physics to medical science.

A periodic annual review of journals and articles guarantees the highest quality of these repositories; this study was chosen thanks to the relevance and quantity of results devoted to different sectors. Thus, the comprehensiveness of these databases allowed us to find studies on circular bioeconomy and business models. The opportunity to use additional sources such as Google Scholar has been discarded due to the minor accuracy of results and difficulties in querying results according to title, abstract, and keyword, which is necessary for using the PRISMA method.

After the dataset identification, elective criteria selection—leading the funnel search process—has been selected. In the first stage, all peer-reviewed articles of the literature from 2014 to 2020 are identified, counted, and reviewed. Megatrends of industrial research are observed between 2014 and 2020 to have a complete view of their impact in triggering the development of circular bio-based business models. With this purpose, the sources querying is conducted through the following keyword combinations: "circular bioeconomy", "bioeconomy", "business model", and "biomass". According to the PRISMA method, the selection process is performed in four steps for the creation of a flowchart:

1. Identification of relevant research searching on different databases through keywords use ($n = 504$). Additional records are identified through research in other sources ($n = 15$).
2. After removing duplicates ($n = 320$), records are screened based on title analysis and abstracts ($n = 294$), removing 26 articles.
3. In the third step, 273 records are chosen to conduct this research on circular bioeconomy and possible business models, removing 21 articles.
4. Finally, the study includes a quantitative and qualitative analysis of the obtained results.

All the systematic literature steps followed the review process shown in Figure 1 and datasets' queries led by keywords combinations reported in Table 1 (Appendix A, Table A1).

**Table 1.** Keyword search terms with databases for systematic review.

| Keywords | Dataset | N° | Dataset | N° | Add Articles | N° |
|---|---|---|---|---|---|---|
| Circular bioeconomy | Scopus | 120 | ScienceDirect | 100 | Other | 12 |
| Circular bioeconomy business model | Scopus | 6 | ScienceDirect | 5 | Other | 0 |
| Bioeconomy business model | Scopus | 12 | ScienceDirect | 5 | Other | 3 |
| Circular bioeconomy business model biomass | Scopus | 0 | ScienceDirect | 1 | Other | 0 |
| Bioeconomy business model biomass | Scopus | 7 | ScienceDirect | 2 | Other | 0 |
| **Total** | | **145** | | **113** | | **15** |

The progressive selection process explores circular bioeconomy business models as the articles' main topic. The identified 519 items are subjected to evaluation by eliminating 225 duplicates in two subsequent steps. A further screening round has allowed eliminating other 21 items, thus reaching the number of 273 eligible articles. The critical evaluation led articles selection, implying the exclusion of those not pertinent to the study, which did not lead to a systematic analysis of circular business models. Finally, 273 items are identified for the qualitative analysis: a historical and geographical analysis, fields of application, and the industrial research trends.

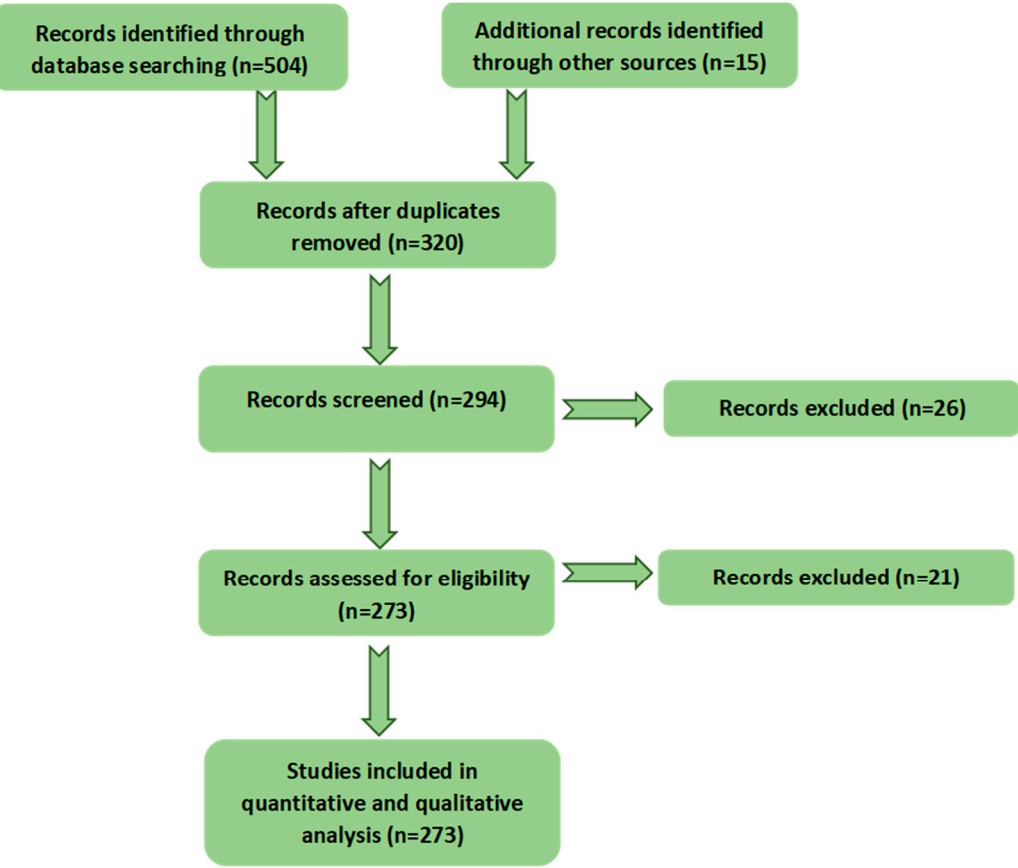

**Figure 1.** Systematic literature review flow diagram from Preferred Reporting Items for Systematic Reviews and Meta-Analyses method (PRISMA) methodology.

### 3.2. Investigation of EIC Accelerator Data Hub

Designed to support SMEs and innovative start-ups on the most critical innovation phase of their growth, the European Innovation Council (EIC) Accelerator Pilot is the new name for what before 2019 was formerly known as the SME Instrument Phase II. It supports top-class innovators to develop and bring to the market new innovative products and business models that could drive economic growth and have the potential to scale up. Opened in 2014, the SME Instrument is part of the EU financial measure of the Horizon 2020 (2014–2020) Industrial Leadership pillar to support ideas with a high-risk and high-potential toward commercialization. Piloting, testing, demonstration, and market replications are all actions supported by the program to boost high-risk and cutting-edge business ideas from the lab to the market.

In the current study, the SME Instrument—Phase II is deemed first among the European funding instruments to investigate megatrends in industrial research on circular bioeconomy business models to overcome the *Valley of Death* and successfully reach the market uptake.

The research analyzes the funded projects registered into the EIC Accelerator Data Hub, which is an interactive digital tool by the Executive Agency for SMEs, generating information (dynamic map, beneficiaries, statistics) on ten EU funding programmes, the Bio-based Industries JU, COSME, EMFF, Horizon 2020 (Energy and Climate action), EIC (Accelerator, Fast Track to Innovation, Pathfinder Open), Innosup, and Maritime. It has proved necessary to understand how many leading companies use the SME Instrument as an accelerator to bridge the *Valley of death*.

Some selection criteria have been set to search through this interactive tool:

- Project Phase (Phase I or Phase II)
- Participant type (Coordinators or Partners)

–     Country and Regions among the 28 Member States and Associated Members
–     Topics among 20 relevant key-enabling technologies and applications fields
–     Budget
–     Call Date (from 2014 to 2020)
–     Project start and end date

The qualitative analysis considers projects funded in Phase II, with TRL ranging from 5 to 6–8 where testing activity concerns a primarily technological innovation that is ready for scalability and placing on the market. The full list of countries and call year provided by the EIC Accelerator Data Hub are part of the data filtering. Agriculture and fisheries, biomarkers and diagnostic medical devices, biotechnology, business model innovation, eco-innovation, and raw materials were chosen as primary topics to investigate innovative funded projects in the bio-based sector.

Data obtained by EIC Accelerator Data Hub are integrated by those filtered in the Community Research and Development Information Service (CORDIS), which is the European Commission's primary source of results from the projects funded by the EU's framework programs for research and innovation (FP1 to Horizon 2020). Based on this analysis, an innovative case study of a circular bioeconomy business model is conducted and analyzed.

### 3.3. Identified Projects' Analysis and Classification

The funded projects selected through the extraction of EIC Accelerator Data Hub constitute case studies to validate circular bio-based business models that have reached the market. The application of a funnel and progressive refinement methodology is complemented by the consultation of data classified in CORDIS, which is the European Commission's primary source for results of EU-funded research and innovation projects. In the present study, the database supports the analysis of bio-based objectives, technologies, business models, and the development status of the project itself (ongoing or finished). Besides, in this case, a funnel-shaped approach has been applied.

It allows screening many projects granted by the SME Instrument program, which were subsequently screened to select those focused on bio-based products development and the associated business models. So, the CORDIS platform's integrated use further narrowed the sample, consisting of completed projects for which information on the technology used and the products obtained is available. Finally, a classification of the projects by strategic orientation "technology-push" and "market-pull" is carried out based on nine different application sectors: eco and bio-based materials, water and wastewater treatments, green chemistry, biogas and biofuels, nutraceuticals, wood and eco-construction, microalgae, biopesticides, and medical.

### 3.4. Validity and Limitations of Research

Data collection for the systematic literature review is restricted to searched specific keyword combinations: *circular bioeconomy*, *bioeconomy*, *business model,* and *biomass*.

All these keywords have synonyms that have not included in the dataset query (i.e., green economy or bio-based economy). However, we have tried to limit the use and combination of keywords as established in Section 3.1 Search and Selection Criteria for systematic literature review. Regarding geographical distribution, data are related to the authors according to their primary affiliations, therefore representing only the countries where the literature is produced.

## 4. Results

A systematic literature review is described and evaluated in qualitative and quantitative terms and considerations about circular bioeconomy topic transformation over time, thematic trends, and outlooks are formulated. Simultaneously, the Investigation of

EIC Accelerator Data Hub supported the exploration of innovative case studies of circular bioeconomy business models funded by the SME Instruments Phase II since 2020.

### 4.1. Analysis of the Evolution of the Literature/Quantitative Analysis

The PRISMA methodology application allows the final identification of 273 publications, belonging in the 2014–2020 period and heterogeneously distributed with a significant increase in literature production from 2018 to 2020 (220), representing 81% globally. Records identified during the systematic literature review and shown in Table 1 have been catalogued according to keyword search terms. Circular bioeconomy represents the most common keyword used in 85% (232) of the publication investigated in the period. Keywords cover the remain 15% are "Bioeconomy business model" (7%), "Circular bioeconomy business model" (4%), "Bioeconomy business model biomass" (3%), and "Circular bioeconomy business model biomass" (1%)—Figure 2.

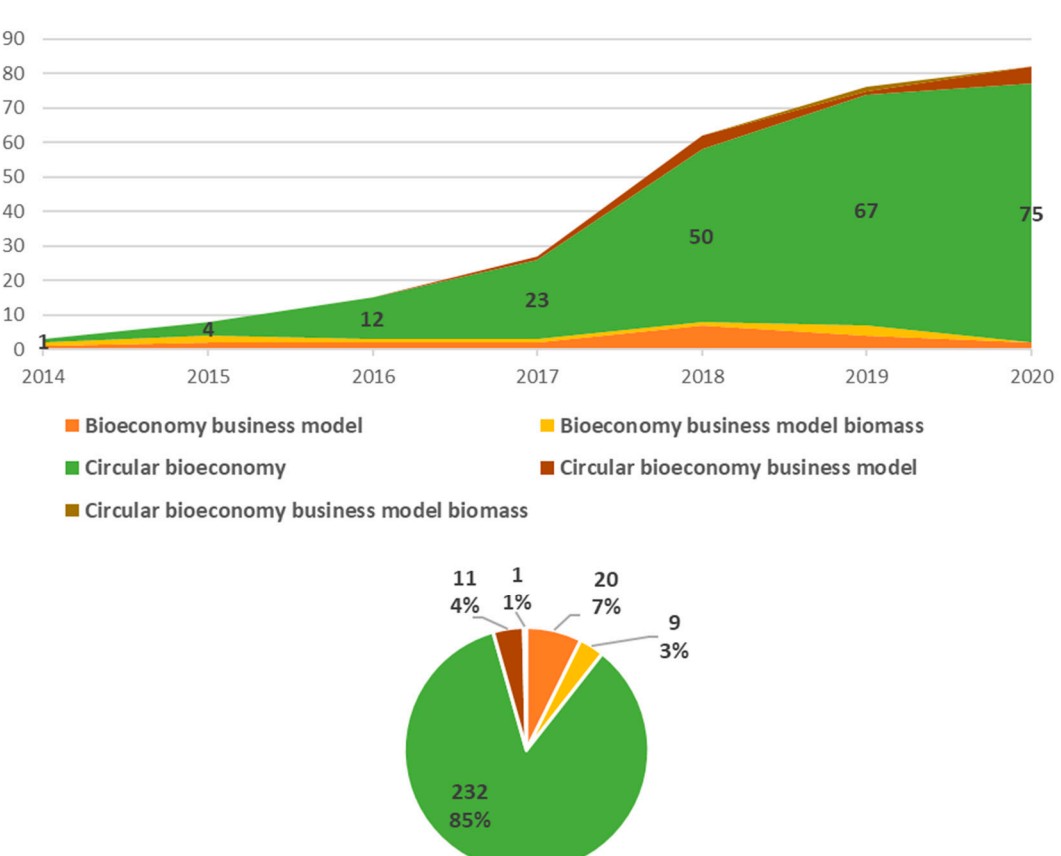

**Figure 2.** Temporal distribution of identified articles and publications by PRISMA methodology and total breakdown by keywords.

Academic literature on bioeconomy and circular bioeconomy thematic areas are border distributed across more than 100 peer-review journals. A ranking of the top journals involved in these topics reveals the Journal of Cleaner Production's primary role, the Bioresource Technology, Sustainability and the Forest Policy, and Economics cover one-third of the literature worldwide.

With a heterogeneous geographical distribution of literature production on circular bioeconomy—based on the first authors' affiliation, 65% of the total is represented by Spain (10.3%, 28), Germany (9.5%, 26) and Finland (9%, 25), followed by India, Italy, Sweden, Greece, Brazil, the UK, and China—as shown in Figure 3.

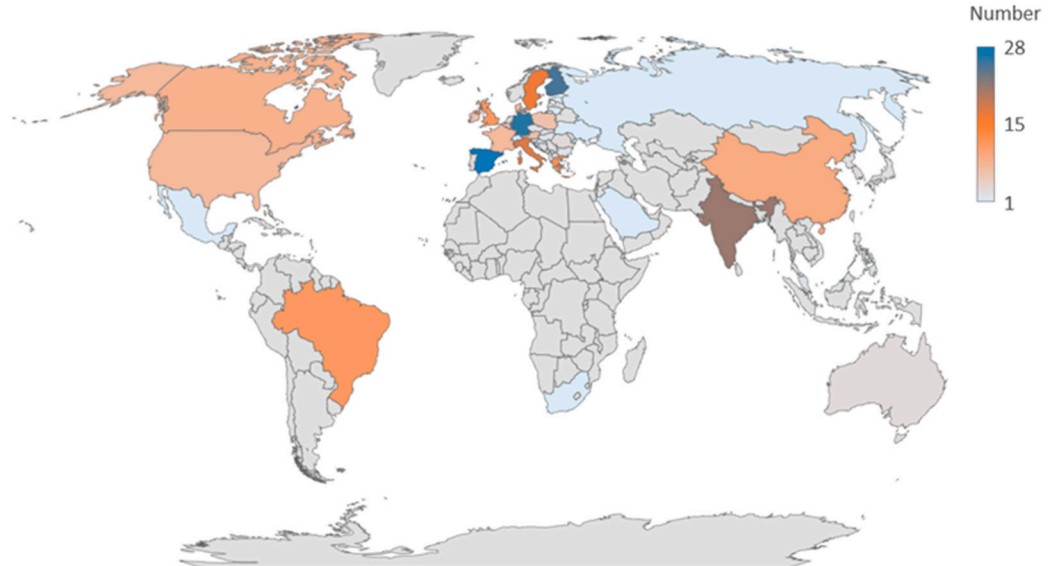

**Figure 3.** Geographical distribution of the reviewed studies according to the affiliation of the authors (number of articles).

Regarding the major institutions involved, there are the Spanish Universities of Madrid, Barcelona, Seville, and the Joint Research Centre-European commission. The Helmholtz Centre for Environmental Research—UFZ from Germany is one of the world's leading research centers in the environmental research field. Finally, in Finland, there are the Universities of Aalto and Helsinki and the Helsinki Institute of Sustainability Science.

*4.2. Qualitative Analysis*

The literature sample obtained through the PRIMA methods is organized according to the predominant thematic domain to explore the business models' characteristic and key success factors investigated—see Table 2.

**Table 2.** Definition of categories.

| Category | Definition |
| --- | --- |
| Agro-Food and Urban Waste Sidestreams. | Valorization of raw materials, agro, and urban waste treatment. |
| Bioenergy and Biofuel. | Production of green energy starting from alternative raw materials. |
| Biopolymer and Bioplastic. | Extraction and transformation processes to obtain bio-based polymers from feedstock and different kind of biomass. |
| Bulk Chemicals or Lignocellulosic Molecules. | Production, in continuous process, organic and inorganic chemicals (i.e., solvents, lubricants, resins, and oil) from row material and valorization lignocellulosic feedstock in a large scale. |
| Fine Chemicals and Pharma. | Development of bio-based products with high added-value, chemicals, ingredients, and cosmetics in small, limited quantities in plants by batch or biotechnological manufacturing processes. |
| Policy, Strategy, and Management. | Analysis and evaluation of policy and strategy proposal; development and implementation of business models and R&D strategies. |

Developing coherent policy and supporting innovation governance represents the most common specialization area of investigation for unlocking the potential of SMEs—see Figure 4.

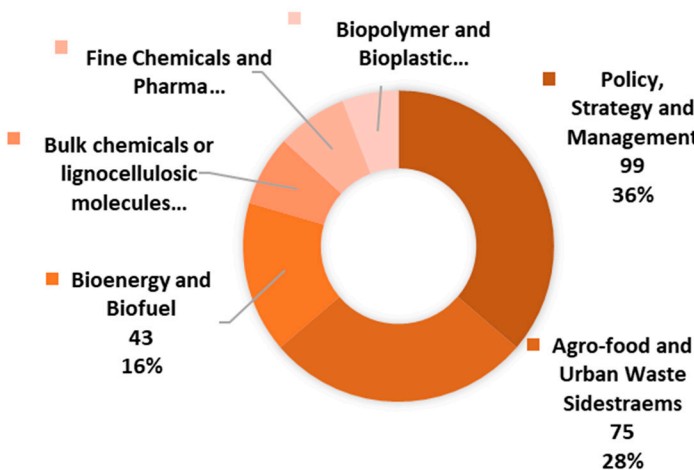

**Figure 4.** Distribution in the leading six categories.

Theoretical studies to policy and innovation management on adopting circular business models and their role in a successful shift to a circular bioeconomy represent over one-third of articles identified (36%).

Agro-food and urban waste sidestreams account for 27% of the articles and regard the use and valorization of feedstocks, biomasses, and nutrients. For instance, it includes agro-industrial wastes, livestock farms, wastewater and Organic Fraction Municipal Solid Waste (OFMSW) to produce starting products with high added value for a positive effect on biodiversity. The development of economic models that provide the integration between the agro-food and bioindustries sectors opens the prospects to develop new potential value chains and provides to generate bio-products with high potential. Bioenergy and biofuel from renewable biological sources are considered in 16% of the articles focused on power, heat, and green biofuels generation. A circular bioeconomy approach applied on green mobility is a key pillar of the Green Deal as the primary tool to reach a carbon-neutral economy sector and increase the absorption of $CO_2$ in biomass, agricultural, and forestry soils. Fine chemicals and pharma thematic areas account for 7% of articles, focusing on producing various fine products in limited quantities in plants by batch or biotechnological manufacturing processes. Using different kinds of processes, they are transformed into chemical compounds, ingredients, and intermediate products for various segments such as pharmaceuticals and cosmetics. Bulk chemicals and lignocellulosic molecules were investigated in 6% of the studies focusing on manufacturing bio-based chemicals from various biomass platforms (i.e., lignocellulosic and agro-based sectors). Finally, 6% of the articles deal with biopolymers and bioplastic, such as the construction of biomaterials and composites, bioplastics polymer, treatment application, and management.

### 4.3. Investigation of the EIC Accelerator Data Hub's Results

The EIC Accelerator Data Hub investigation leads to selecting 265 projects funded under the SMEs Instrument Phase II from 2014 to 2020. Projects aim to reach the market through testing, prototyping, piloting, scaling up, miniaturization, design, and market replication, with a starting TRL equal to or greater than 6. Projects are divided into five technological domains, among which Eco-innovation and raw materials represent 33% of the total (88 projects), followed by Agriculture and fisheries (27%, 71), Biotechnology (19%, 52), Biomarkers and diagnostic medical devices (12%, 31), and Business model innovation (9%, 23). The Work Programme 2016–2017 is the most participated during the period, funding 40% of the total projects and the eco-innovation ones. A diagram of funded projects in the five bio-based areas over the period is reported below—see Figure 5.

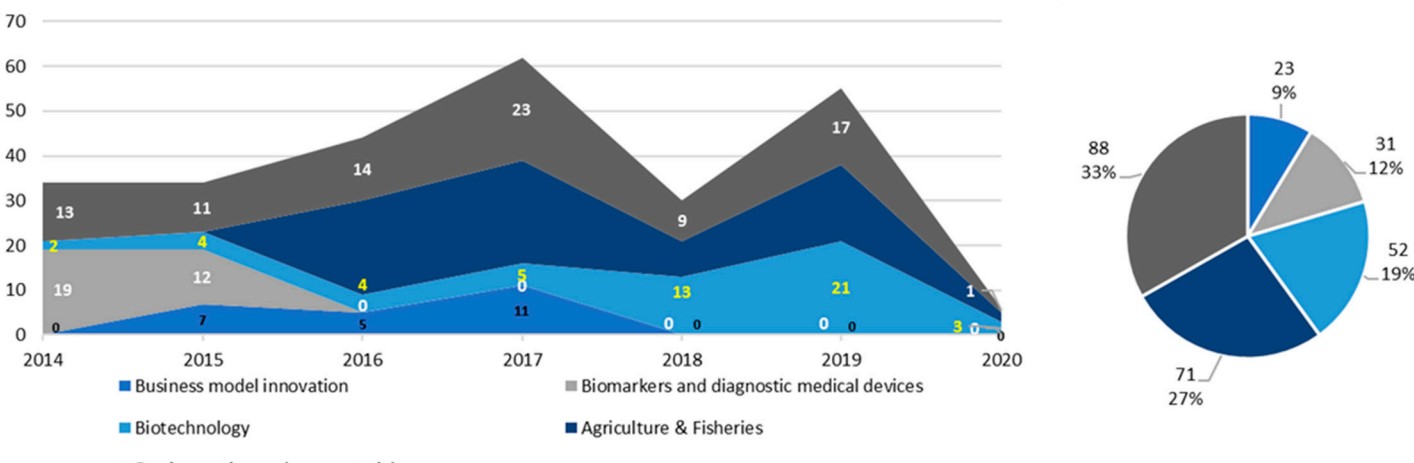

**Figure 5.** Funded projects by the small and medium-sized enterprise (SME) Instruments—Phase II from 2014 to 2020 by five thematic domains.

With a total investment of 457.9 million euros in the 2014–2020 period, projects in the "Eco-innovation" area cover 29% of the allocated funds (131.3 million euros ($1.31 \times 10^8$)), followed by "Biomarkers and medical-diagnostic devices" and "Agriculture and fisheries" (22%, each equal to 102and 100.4 million euros ($1.02 \times 10^8$ and $1 \times 10^8$)), "Biotechnology" (21%, 94.5 million euros ($9.46 \times 10^7$)), and "Business model innovation" (6%, 29.6 million euros ($2.9 \times 10^7$)).

Projects belonging in the "Eco-innovation" area mobilized 188.5 million euros of investments, of which European funds constitute around 70% (131.2 million euro ($1.3131 \times 10^8$)). In the case of SME Instrument 2016–2017 Phase II, proposals can request an EU contribution between 1 and 5 million euro, which is equal to 70% of the budget, and exceptionally, 100% where the research component is strongly present—see Figure 6.

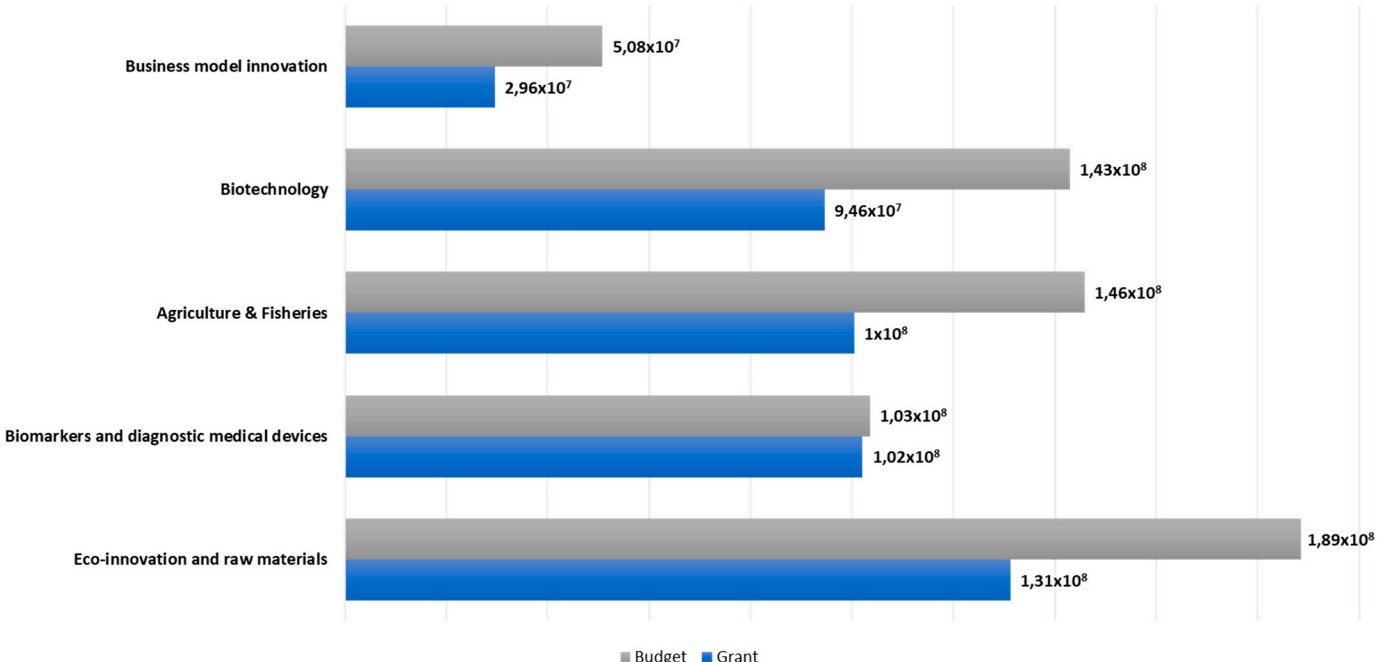

**Figure 6.** Budget vs. grant rate per each of five thematic areas of projects funded under the SME Instrument—Phase II.

*4.4. Analytical Evaluation of Identified Projects*

The EIC Accelerator Data Hub survey made it possible to identify a total of 265 projects, which were subsequently subjected to manual screening to determine circular bio-based business models. This funnel-shaped approach has made it possible to select progressive refinements of the completed projects for which information on the used technology and the products obtained is already available (published on the CORDIS platform). A group of 48 case studies were then divided according to the two strategic orientations "**technology-push**" (31) and "**market-pull**" (17)—Table 3.

**Table 3.** Case studies.

| Orientations | Year | Call | Project Acronym | Application Sector |
|---|---|---|---|---|
| **MARKET PULL** | 2018 | H2020-SMEInst-2018-2020-2 | REBICOM | Eco and bio-based materials |
| **MARKET PULL** | 2018 | H2020-SMEInst-2018-2020-2 | Simecos | Nutraceuticals |
| **MARKET PULL** | 2017 | H2020-SMEINST-2-2016-2017 | BioAXOS | Nutraceuticals |
| **MARKET PULL** | 2017 | H2020-SMEINST-2-2016-2017 | ecoSave | Eco and bio-based materials |
| **MARKET PULL** | 2017 | H2020-SMEINST-2-2016-2017 | Green-linker | Green chemistry |
| **MARKET PULL** | 2017 | H2020-SMEINST-2-2016-2017 | HOMEBIOGAS | Biogas and biofuels |
| **MARKET PULL** | 2017 | H2020-SMEINST-2-2016-2017 | HYDROBLOOD | Nutraceuticals |
| **MARKET PULL** | 2017 | H2020-SMEINST-2-2016-2017 | INNOPREFAT | Nutraceuticals |
| **MARKET PULL** | 2017 | H2020-SMEINST-2-2016-2017 | SOLARIS | Biogas and biofuels |
| **MARKET PULL** | 2017 | H2020-SMEINST-2-2016-2017 | WineLeather | Eco and bio-based materials |
| **MARKET PULL** | 2016 | H2020-SMEINST-2-2016-2017 | INTERCOME | Microalgae |
| **MARKET PULL** | 2016 | H2020-SMEINST-2-2016-2017 | LIFEOMEGA | Nutraceuticals |
| **MARKET PULL** | 2016 | H2020-SMEINST-2-2016-2017 | PAPTIC | Eco and bio-based materials |
| **MARKET PULL** | 2015 | H2020-SMEINST-2-2015 | BIOCURE | Eco and bio-based materials |
| **MARKET PULL** | 2015 | H2020-SMEINST-2-2015 | IcoCell | Medical |
| **MARKET PULL** | 2014 | H2020-SMEINST-2-2014 | ADD-ON | Biogas and biofuels |
| **MARKET PULL** | 2014 | H2020-SMEINST-2-2014 | BLOSTER | Biopesticides |
| **TECHNOLOGY PUSH** | 2014 | H2020-SMEINST-2-2014 | APEX | Green chemistry |
| **TECHNOLOGY PUSH** | 2015 | H2020-SMEINST-2-2015 | CleanOil | Eco and bio-based materials |
| **TECHNOLOGY PUSH** | 2016 | H2020-SMEINST-2-2016-2017 | CLEANTECH-BLOCK2 | Wood and eco-construction |
| **TECHNOLOGY PUSH** | 2014 | H2020-SMEINST-2-2014 | CLIPP PLUS | Eco and bio-based materials |
| **TECHNOLOGY PUSH** | 2016 | H2020-SMEINST-2-2016-2017 | CO2Catalyst | Green chemistry |
| **TECHNOLOGY PUSH** | 2016 | H2020-SMEINST-2-2016-2017 | DEPURGAN | Biogas and biofuels |
| **TECHNOLOGY PUSH** | 2017 | H2020-SMEINST-2-2016-2017 | ECOSHEET-PRO | Eco and bio-based materials |
| **TECHNOLOGY PUSH** | 2014 | H2020-SMEINST-2-2014 | ECO-SILENTWOOD | Wood and eco-construction |
| **TECHNOLOGY PUSH** | 2015 | H2020-SMEINST-2-2015 | H2AD-aFDPI | Water and wastewaters treatments |
| **TECHNOLOGY PUSH** | 2015 | H2020-SMEINST-2-2015 | HTC4WASTE | Water and wastewaters treatments |
| **TECHNOLOGY PUSH** | 2018 | H2020-SMEInst-2018-2020-2 | HTCycle | Water and wastewaters treatments |
| **TECHNOLOGY PUSH** | 2016 | H2020-SMEINST-2-2016-2017 | INDALG | Microalgae |
| **TECHNOLOGY PUSH** | 2015 | H2020-SMEINST-2-2015 | InnoPellet | Biogas and biofuels |
| **TECHNOLOGY PUSH** | 2014 | H2020-SMEINST-2-2014 | iPURXL | Water and wastewaters treatments |
| **TECHNOLOGY PUSH** | 2015 | H2020-SMEINST-2-2015 | Lt-AD | Water and wastewaters treatments |
| **TECHNOLOGY PUSH** | 2017 | H2020-SMEINST-2-2016-2017 | MOSSWOOL | Eco and bio-based materials |
| **TECHNOLOGY PUSH** | 2017 | H2020-SMEINST-2-2016-2017 | MUBIC | Biogas and biofuels |
| **TECHNOLOGY PUSH** | 2016 | H2020-SMEINST-2-2016-2017 | nanoHPcs | Eco and bio-based materials |
| **TECHNOLOGY PUSH** | 2017 | H2020-SMEINST-2-2016-2017 | PFS | Water and wastewaters treatments |
| **TECHNOLOGY PUSH** | 2016 | H2020-SMEINST-2-2016-2017 | PHOSave | Green chemistry |
| **TECHNOLOGY PUSH** | 2014 | H2020-SMEINST-2-2014 | PlugBioIn | Green chemistry |
| **TECHNOLOGY PUSH** | 2017 | H2020-SMEINST-2-2016-2017 | reNEW | Water and wastewaters treatments |
| **TECHNOLOGY PUSH** | 2016 | H2020-SMEINST-2-2016-2017 | REW-TYRES | Eco and bio-based materials |
| **TECHNOLOGY PUSH** | 2018 | H2020-SMEInst-2018-2020-2 | Rosalind | Green chemistry |
| **TECHNOLOGY PUSH** | 2017 | H2020-SMEINST-2-2016-2017 | sFilm-FS | Eco and bio-based materials |
| **TECHNOLOGY PUSH** | 2016 | H2020-SMEINST-2-2016-2017 | SMARTSAND | Wood and eco-construction |
| **TECHNOLOGY PUSH** | 2015 | H2020-SMEINST-2-2015 | WATLY | Water and wastewaters treatments |

| TECHNOLOGY PUSH | 2015 | H2020-SMEINST-2-2015 | WHEY2VALUE | Green chemistry |
| TECHNOLOGY PUSH | 2014 | H2020-SMEINST-2-2014 | WINTHERWAX | Wood and eco-construction |
| TECHNOLOGY PUSH | 2018 | H2020-SMEInst-2018-2020-2 | Woodoo | Eco and bio-based materials |

The selection and classification have been carried out manually for each project and previously through the data available on CORDIS, which are essential for analyzing each project's purpose, themes, and technologies. The business models' qualitative study is supported by an extensive market analysis of the selected applications' current state and growth forecasts and their belonging into the two strategic orientations.

Table 3 shows the most relevant project's information, such as orientation, start year, type of call, project acronym, and application sector: Eco and bio-based materials (13 projects), Water and wastewater treatment (9), Green chemistry (7), Biogas and biofuels (6), Nutraceuticals (5), Wood and eco-construction (4), Microalgae (3), Biopesticides (1), and Med (1).

In this context, thanks to bibliographic analysis, the main bio-based products that manage to overcome the Valley of Death barrier are, in most cases, projects that respond to market demands. Among them, projects belonging in Eco and bio-based materials (5) and Nutraceuticals (5) applications are the most represented, followed by Biogas and biofuels (3), Microalgae (2), Green chemistry (1), Biopesticides (1), and Med (1).

## 5. Discussion

Bioeconomy and circular economy concepts came into being between the 1970s and the 1990s [40] and have triggered increasingly prominent studies, thanks to a growing demand for new products and innovative solutions with low environmental impact. In Europe, policy and financial measures to facilitate the transition from a linear to a circular economy have boosted new industrial opportunities, particularly for SMEs. The most recent definition of circular bioeconomy opens the discussion for a new era based on the sustainable consumption of biological and renewable resources as an essential and innovative solution for sustainable development.

However, despite the high potential, technologies and solutions deriving from biomass and circular processes are still an emerging sector, which is characterized by dynamic research even though a limited number of products are on the market. The trade-off between biomass use and the profitable manufacturing scale-up plays a crucial role in supporting adopting circular bioeconomy business models and developing products and/or services with high added value. Innovations based on the exploitation of biomass and renewable biological resources are research-intensive, finding primary financing sources in grants and public subsidies. However, the management of scale-up in overcoming the so-called "*Valley of Death*" is the most capital-demand phase in the business model validation. Technical and operational barriers, lack of private investors' confidence, unfavourable demand, and regulatory conditions are among the factors that hinder the placing of bio-based solutions on the market.

This section contains all the findings and their implications linked to the two methodologies adopted in the article: systematic analysis of the publications by the PRISMA method (systematic literature review) and the EIC Accelerator Data Hub investigation. According to the authors' affiliation, the first method reported the temporal distribution of identified articles and publications, especially the geographical distribution of the reviewed studies. While the second methodology applied in this study, the EIC Accelerator Data Hub analysis focuses on a sample of 48 projects funded by SME Instruments—Phase 2. A classification of the projects is carried out based on the "*technology-push*" and "*market-pull*" strategic orientation based on nine different application sectors: Eco and bio-based materials, Water and wastewater treatment, Green chemistry, Biogas and biofuels, Nutraceuticals, Wood and eco-construction, Microalgae, Biopesticides, and Medicines. The comparative analysis of projects with circular business models has allowed defining the crucial elements for a bio-based product's success overcoming the *Valley of Death*. In

detail, in the final part of the discussion section, all nine application sectors have been deepened, and for each of them, additional in-depth analysis is provided.

Emerging studies have contributed to the enrichment of circular bioeconomy litera­ture and the development of sustainable business models. A substantial increase in pub­lications and articles in the last three years (from 2018 to 2020) has occurred from a first qualitative analysis, which was also fostered by the European policy and regulatory initi­atives in support of the bioeconomy and the circular economy. The European Bioeconomy Strategy last updated in 2018 and the New Green Deal launched in 2019 represent the last and most significant contribution to boost sustainable economic growth in achieving the Sustainable Development Goals (SDGs), mobilizing over 100 billion euros in investments for the carbon-neutral transition.

According to the systematic literature review, the most productive countries in stud­ies on bioeconomy circular business models are primarily European (Spain, Germany, Finland, Italy, Sweden, Greece), representing 45% of the overall production, followed by China, the UK, India, and Brazil belonging in the top ten authors. Spain is the country where more authors have published on the circular bioeconomy, which is fostered by the decision taken in 2014 to design a specific bioeconomy strategy that was launched in Jan­uary 2016 to improve bioeconomy based on the application of bio-based products. The Spanish Strategy addresses the global challenges related to the circular bioeconomy, boosting interlinks between agricultural and biotechnological sciences and the agri-food, biotech, and biomass industrial sectors [41]. Most of the articles are focused on exploiting biomass, biowaste, and agro-food waste from the agro-industry as promising feedstocks for advanced manufacturing based on third-generation biorefineries [42]. They encom­pass a wide range of systems and steps, referred to a single biomass conversion process or a complex plant of integrated poly-generation with other industries using green chem­ical transformations, biochemistry, and thermochemistry approach [43].

Germany and Finland represent the second and third countries with the highest number of publications in circular bioeconomy business models and encouraged by a strong national legislative roadmap. In 2010, Germany published its first national Re­search Bioeconomy Strategy with more than €2.4 billion allocated [44]. The Finland bio­conomy strategy launched in 2014 aims to increase the yield of the bioeconomy to 100 billion euros and create 100,000 new bioeconomy jobs by 2025. There is a common direc­tion in both countries: enhance the forestry sector and, therefore, forestry policies to sup­port the development of high value-added products from this starting feedstock [45,46]. The predominant circular bioeconomy business models in Finland and Germany focus on producing biopolymers, fibers, and construction materials, including considerations on sustainability along the production life cycle (LCA) and nutrients recovery adopting a cascade approach.

Placing on the market and the raising of financial capital are among the main chal­lenges for SMEs, which are related to "*the lack of financial resources, technology, inadequate information systems […]*" and human-based barriers, such as "*company leadership or the lack of customer interest in the environment*" (Ormazabal et al. 2018, p. 166) [47]. The absence of financial resources is a typical phenomenon observed in the product-innovation literature as the "*Valley of Death*", where resources are more easily found during the R&D (Research and Development) process than during the commercialization phase [48]. Moreover, busi­ness models in the bioeconomy sector require a considerable volume of investments with high operational and technological risks responsible for limited private investors' confi­dence. The support of venture capitals is also connected to the strategic **"technology-push"** and **"market-pull"** orientation of the business model, which according to real or presumed market demand affects the ability to ensure stable long-term revenues.

In this study, the assessment of **crucial success factors in overcoming the "*Valley of Death*"** of circular bioeconomy business models is based on a systematic analysis of a sample of **48 projects** funded by SME Instruments—Phase 2. They encompass test and

demonstration activities of technological innovations ready for scalability and commercialisation with a TRL between 5 and 6–8.

The comparative analysis of the business models allowed the definition of the crucial elements for the placing on the market of a bio-based product. Flexible solutions aimed at the partial or total replacement of fossil-based products belonging to "market-pull" orientation have successfully overcome the *Valley of Death*. Optimal essential factors are found in models aimed at improving the production process, limiting emissions, and minimizing material losses. Solutions ensure a biomass supply compatible with market demand, with high yields and a consequent increase in product competitiveness and the process's characteristics.

Therefore, a ranking of circular bioeconomy business models with the highest competitive potential, not only in terms of innovation but also for acceptability, has been carried out. Successful cases belong to *eco-sustainable/bio-based materials*, *nutraceutical* (pet and human food), and *microalgae* sectors.

*Eco-sustainable and bio-based materials*. In recent years, growing global awareness that different climate changes and related consequences are closely associated with human activities is taking hold. Therefore, the offer is adapting to the consumers' lifestyle trends that increasingly consider the possibility of improving the global situation through eco-sustainable behaviour, including consuming bio-based materials. The global bio-based materials market has been estimated at approximately 11,000.0 million USD in 2017 and is expected to generate revenue of 94,150.0 million USD by the end of 2026 [49]. The bioplastics sector is also part of this scenario, whose market is currently worth 6.04 billion USD and is expected to reach 19.93 billion USD in 2026 [33]. Bioplastics represented about 2% of the total plastics market in 2015, and they will grow to 40% by 2030, mobilizing employees from the 23,000 jobs to 300,000 by 2030 [50].

In the present study, several business models for the production of biopolymers and biomaterials from biomass by-products have been investigated to identify the crucial factor of their success on the market. The production of biomaterials, biopolymers, and bioplastics for packaging, textiles, and coatings are predominant. Among the significant example, the WineLeather project designed a biopolymer obtained from wine industry waste (grape marc) to produce organic and cruelty-free leather. The project aims to enhance the wine industry's biowaste and develop a more sustainable process, proposing alternative synthetic leather to minimize the use of chemicals, water, and waste produced and avoid VOCs emissions (Volatile Organic Compounds). The owner SME, pushed toward new commercial opportunity, serves a multi-product market: packaging, clothing, and automotive. The REBICOM project develops a biodegradable and compostable recyclable film that is better in weight, density, and production costs than competing bioplastics. It is based on an enzymatic technology complex of nine main ingredients derived from plant by-products, which become biodegradable and compostable under bacterial biological activity. The project introduces innovative technology for packaging and laminating films applications complying with the European compostability and biodegradability standards. Meeting the demand for ecological food packaging whose global value was approximately 178.6 billion USD in 2019 and a forecast of 246.3 billion USD by 2025 [51], the PAPTIC project proposes a new wood fiber-based material that combines the renewability of paper with efficiency and functionality plastic's resource. In this field, the EcoSave Packaging (ESP) project brings its innovative packaging material to the market by an owned assembly process that successfully responds to all social, economic, and environmental needs. EcoSave Packaging has developed a resource-efficient, environmentally friendly, high-quality, and differentiated packaging patented concept that will radically impact the food packaging industry.

Within the macro-sector of eco and bio-based products, there has been an increase in relevance for biomaterials, which are designed to interact with biological systems for medical purposes, whether diagnostic or therapeutic. The global market value of biomaterials was 83.76 billion USD in 2017, and it is forecasted to increase to nearly 152 billion USD in

2021 [52]. The BIOCURE project aims to develop a new biomaterial-based and cost-effective wound dressing based on a novel biomaterial derived from the eggshell membrane to be used as first-line treatment in all wounds at risk of delayed or non-healing. The global market for advanced wound care products was projected to be 3 billion USD in 2012, with annual growth above 10% and representing one of the leading medical product sectors. EggShell Membrane-Based Wound Dressing (ESM) meets both the healthcare sector's needs and the market's needs because they cost significantly less (cost at least three times lower), are much easier to scale, and are ultimately considerably safer. The combination of scalability, effectiveness, and price of the product represents a breakthrough in the wound-healing market.

*Nutraceutical*. Globally, nutraceuticals are gaining importance and are becoming a part of the consumer's daily diet. In 2017, the global nutraceutical market was worth approximately 383 billion USD in 2017, and it is expected to reach 561.4 billion USD by 2023 [53]. Among the most promising substances, the protein ingredients market size was 38.02 billion USD in 2019, with an annual growth rate of 9.1% to 2027, followed by other dietary supplements such probiotics, prebiotics, flavonoids, flavones, carotenoids, beta carotene, and omega−3.

In the present study, a successful business model-based circular approach has been analyzed, such as the BioAXOS project dealing with the production of a new effective and affordable prebiotic from the XOS family that can replace inulin in at least half of its applications. The project aims to bring highly effective and affordable prebiotic soluble fiber ingredients for pet food, human food, and nutraceutical applications to the market. It is a selected form of arabinoxylans dietary fiber derived from corn known to intestinal health experts for its prebiotic potential not yet commercially available. Gaining a market share of at least 20%, in Europe, it represents a market of 200 million euros. Therefore, it represents a valid alternative to biomass use and considers the growing market demands in this specific sector. Some nutraceuticals can be used as essential nutrient co-adjuvant cancers (breast, lung, and pancreatic). Examples are the two projects: LIFE OMEGA and Simecom. The first aims to produce a high concentration nutritional product of Omega3 EPA improving cancer patients' health; the second, a product with an anti-inflammatory effect, is based on the enzyme chitinase YKL−40 inhibition. Both projects address the market's needs and demands, trying to respond to the urgent need for better treatments to increase these patients' life expectancy.

*Microalgae*. Microalgae can generate many compounds, such as vitamins, proteins with essential amino acids, polysaccharides, fatty acids, sterols, pigments, fibers, and enzymes with superior characteristics to the corresponding synthetic compounds [54,55]. Thanks to numerous applications potential in bioenergy, pharmaceutical, and nutraceutical fields, microalgae represent a rapidly growing sector. The global microalgae derivatives market was worth 2.36 billion USD in 2019, and it is expected to reach 3.9 billion USD by 2027 [56].

In the present study, the INTERCOME project represents a valuable success business model to obtain products derived from microalgae biomass for agriculture, nutrition, cosmetics, and aquaculture applications. The INTERCOM project promoted the algae photosynthesis optimization to increase the production of valuable products from algal biomass. Project partners designed and implemented an existing production facility and developed an operational protocol to cultivate various microalgae strains without risking contamination. The research team increased the selected crop strains' volume for four main products: agricultural biostimulants, dermo-cosmetics for skincare, nutritional additives, and feed.

Circular bioeconomy business models belonging in the "technology-push" strategic orientation struggle to reach the market due to multiple barriers and obstacles. In the present study, projects in water and wastewater treatment sectors, despite highly innovative solutions, require more significant investments to face technical and operational challenges and a financial requirement to support the demonstrative phase. New approaches

have been identified to tackle water and sludge treatment, guaranteeing efficiency and environmental sustainability processes. In this field, the reNEW project aims to validate technology for turning wastewater into high-value products by recovering essential acids and nutrients, while the PFS and iPURXL projects pursue reducing the concentrations of pollutants, toxic substances, and pharmaceutical products through water purification solutions. Although the sector is continuously expanding, these products are not yet placed on the market and require a feasibility plan to integrate into current water treatment and purification systems and market acceptability.

The biomass supply chain's sustainability and efficiency are among the most significant obstacles for bioenergy and biorefineries production in the demonstration phase. In this study, most of the projects from green chemistry, bioenergy, bio-based products for agrochemical, eco-construction, and medical applications have shown a disruptive industry potential. They have validated innovative processes for products obtained by biorefinery approaches, replace toxic and high environmental impact substances with sustainable alternatives, reduce $CO_2$ emissions, and optimize resources through cascade approaches. However, the crucial factors for developing large-scale processes require abundant and constant raw materials to overcome the seasonal and logistical conditions of biomass transportation and processing, without competing with the food and feed chain. The need to ensure high performance at competitive prices compared to a fossil-based solution implies a biomass supply chain that considers the feedstock characteristics and distribution infrastructure to the final consumer. These latter factors are particularly crucial for biofuels such as biomethane obtained from biowaste of agricultural origin through biogas purification. In Europe, it grew exponentially, increasing form 29.2 petajoules in 2000 to 649.8 petajoules in 2019 [57]. The absence of an integrated logistics system and a robust biomass market are responsible for a slowed capillary development of conversion and distribution plants.

The financial viability of these circular bioeconomy business models depends on multiple factors, such as the market acceptability, the presence of an integrated supply chain that knows how to combine food, feed, fuels from crops, and biofuels production, and regulatory and financial incentives. The models currently emerging on the market are those in which the public leverage of subsidies stimulates joint ventures between farmers and logistic brand owners, leading to greater confidence in entrepreneurs and private lenders.

## 6. Conclusions

This study shows the current situation on the state-of-the-art circular economy, supporting megatrends' definition and the outlook on circular bioeconomy business models. The expansion and evolution of the conceptualization of sustainable circular bioeconomy and associated business models are based on the systematic literature analysis and the EIC Accelerator Data Hub investigation. The results, coupled with literature research, offer insights about challenges and opportunities posed by a transition toward a circular bioeconomy, especially in overcoming the *Valley of Death* through the interaction of "*technology push*" and "*market pull*" strategic orientations.

The SMEs Instruments—Phase II has actively promoted the bio-based sector's unlocking potential, supporting over 459.5 million euros in 265 projects belonging to the main thematic priorities related to using renewable biological resources biotechnology methods.

In more general terms, according to the Horizon 2020 Interactive Dashboard, the SMEs participation on advanced materials, biotechnology, climate action, and bioeconomy thematic priorities results in 1620 H2020 signed grants and 1.44 billion euros of EU Net contribution. Thanks to the EU contribution, disruptive innovations boost new growth perspectives for those bio-based business models based on a circular approach. Results obtained by the Horizon 2020 play a crucial role on setting the scene for the 9FT

Horizon Europe (2021–2027), where stimulating breakthrough, market-creating innovation notably by SMEs, and venture capital investments are the primary drivers [58].

The innovative circular business models selected from the identified projects, particularly all those that fall in the *market pull* strategic orientation, were also functional to identify the main trends in the bioeconomy sector. In summary, the analysis demonstrates that business models successfully overcoming the *Valley of Death* are focused on eco-sustainable/bio-based materials, nutraceuticals, and microalgae production [59]. Versatile bioproducts fit many applications, such as packaging, textile, construction, or medical devices, as well as advances in the nutraceuticals and microalgae sector, which is dominated by the growing demand of probiotics, prebiotics, omega-3, and other supplements and bioenergy.

These products represent the current leading trends, showing greater acceptance by the market acceptability and trust of public and private funding providers. Limiting emissions, reducing material losses, and preserving the environment are among the main challenges facing Europe and the whole world today. Following these considerations, the new Green Deal plays a pillar role in driving SMEs and the academic community to increase their effort in the market validation on circular bioeconomy business models, especially in biomaterials and sustainable mobility as the main drivers toward a carbon-neutral economy.

Although some limitations must be taken into account including the subjectivity of the authors in the bibliographic classification and the limited number of analyzed projects, this article not only provides an updated qualitative and quantitative analysis of the literature produced on business models circulars but also highlights the trends in this driving sector linked to the SME Instrument financial support. Therefore, it represents a starting point for future research that aims to overcome the famous concept of the Valley of Death by promoting the development of new circular business models that consider the market demand.

**Author Contributions:** Conceptualization: F.G. and I.R.; Methodology: F.G.; Analysis of Results: F.G. and I.R.; Writing—Original Draft: F.G. and I.R.; Writing—Review and Editing: F.G. and I.R.; Visualization: F.G. and I.R. All authors have read and agreed to the published version of the manuscript.

**Funding:** The study has been supported by the Bioeconomy Pilot of the Vanguard initiative and the project AlpLinkBioEco (563).

**Institutional Review Board Statement:** Not applicable.

**Informed Consent Statement:** Not applicable.

**Data Availability Statement:** Data is contained within the article or supplementary material. The data presented in this study are available in the Appendix A.

**Acknowledgments:** The study has been carried out in the context of the AlpLinkBioEco (563) and ARDIA-Net (821) projects supported by the Interreg Alpine Space funding programme. Authors thank the Lombardy Green Chemistry Association to provide support through market data and mapping promising business models produced by the Vanguard initiative's Bioeconomy Pilot.

**Conflicts of Interest:** The authors declare no conflict of interest.

# Appendix A

**Table A1.** Publications by datasets and keywords.

| Dataset | Keywords | Title | Authors and Year of Publication |
|---|---|---|---|
| Scopus | **Circular bioeconomy business model** | A new circular business model typology for creating value from agro-waste | Donner M. et al., 2020 |
| | | The management of agricultural waste biomass in the framework of circular economy and bioeconomy: An opportunity for greenhouse agriculture in Southeast Spain | Duque-Acevedo M. et al., 2020 |
| | | Circular economy & sharing collaborative economy principles: A case study conducted in wood-based sector | Pirc Barči A. et al., 2019 |
| | | A systematic literature review of bio, green and circular economy trends in publications in the field of economics and business management | Gregorio V.F. et al., 2018 |
| | | Bioeconomy opportunities in the Danube region | Gyalai-Korpos M. et al., 2018 |
| | | Sustainable business modeling of circular agriculture production: Case study of circular bioeconomy | Ryabchenko O. et al., 2017 |
| Scopus | **Bioeconomy business model biomass** | Potential trade-offs of employing perennial biomass crops for the bioeconomy in the EU by 2050: Impacts on agricultural markets in the EU and the world | Choi, H.S. et al., 2019 |
| | | A systematic approach to exploring the role of primary sector in the development of Estonian bioeconomy | Mõtte M. et al., 2019 |
| | | Stakeholder assessment of the feasibility of poplar as a biomass feedstock and ecosystem services provider in Southwestern Washington, USA | Hart N.M., 2018 |
| | | Context Matters—Using an Agent-Based Model to Investigate the Influence of Market Context on the Supply of Local Biomass for Anaerobic Digestion | Mertens A. et al., 2016 |
| | | Long-Term Yields of Switchgrass, Giant Reed, and Miscanthus in the Mediterranean Basin | Alexopoulou E. et al., 2015 |
| | | Bioeconomy and the future of food—Ethical questions | Kröber B. et al., 2015 |
| | | A spatially explicit techno-economic assessment of green biorefinery concepts | Höltinger S. et al., 2014 |
| Scopus | **Bioeconomy business model** | Marine Bioresource Development—Stakeholder's Challenges, Implementable Actions, and Business Models | Collins J.E. et al., 2020 |
| | | Servitization and bioeconomy transitions: Insights on prefabricated wooden elements supply networks | Pelli P. et al., 2020 |
| | | Bioeconomy development and using of intellectual capital for the creation of competitive advantages by SMEs in the field of biotechnology | Gârdan D.A. et al., 2018 |
| | | The role of public subsidies for efficiency and environmental adaptation of farming: A multi-layered business model based on functional foods and rural women | Varela-Candamio L. et al., 2018 |
| | | Towards a sustainable innovation system for the German wood-based bioeconomy: Implications for policy design | Purkus A. et al., 2018 |

| | | Sustainability-driven new business models in wood construction towards 2030 | Toppinen A. et al., 2018 |
|---|---|---|---|
| | | The influence of intangible assets on the new economy at European level | Irina C., 2018 |
| | | Services in the forest-based bioeconomy—analysis of European strategies | Pelli P. et al., 2017 |
| | | Biorefinery strategies: exploring approaches to developing forest-based biorefinery activities in British Columbia and Ontario, Canada | Blair M.J. et al., 2017 |
| | | Price trends and volatility scenarios for designing forest sector transformation | Lochhead K. et al., 2016 |
| | | Responding to the bioeconomy: Business model innovation in the forest sector | Hansen, E., 2016 |
| | | Investment into the future of microbial resources: Culture collection funding models and BRC business plans for Biological Resource Centres | Smith D. et al., 2014 |
| **Scopus** | **Circular bioeconomy** | An urgent call for circular economy advocates to acknowledge its limitations in conserving biodiversity | Buchmann-Duck J et al., 2020 |
| | | Agricultural waste: Review of the evolution, approaches and perspectives on alternative uses | Duque-Acevedo M. et al., 2020 |
| | | The circular bioeconomy: Its elements and role in European bioeconomy clusters | Stegmann P. et al., 2020 |
| | | Sequential Carotenoids Extraction and Biodiesel Production from Rhodosporidium toruloides NCYC 921 Biomass | Passarinho P.C. et al., 2020 |
| | | Multi-objective optimal synthesis of algal biorefineries toward a sustainable circular bioeconomy | Solis C.A., 2020 |
| | | The contribution of sustainable development goals and forest-related indicators to national bioeconomy progress monitoring | Linser S. et al., 2020 |
| | | Microbial electrosynthesis from $CO_2$: Challenges, opportunities and perspectives in the context of circular bioeconomy | Bian B. et al., 2020 |
| | | The Case for a Lemon Bioeconomy | Ciriminna R. et al., 2020 |
| | | Bioelectrochemical systems for a circular bioeconomy | Jung S. et al., 2020 |
| | | Italy's nutraceutical industry: a process and bioeconomy perspective into a key area of the global economy | Pagliaro M., 2020 |
| | | Food and Non-food biomass production, processing and use in sub-Saharan Africa: Towards a regional bioeconomy | Callo-Concha D. et al., 2020 |
| | | Engineering aspects of hydrothermal pretreatment: From batch to continuous operation, scale-up and pilot reactor under biorefinery concept | Ruiz H.A. et al., 2020 |
| | | Perspectives on "game changer" global challenges for sustainable 21st century: Plant-based diet, unavoidable food waste biorefining, and circular economy | Sadhukhan J. et al., 2020 |
| | | Hybrid life cycle assessment of agro-industrial wastewater valorisation | Chen W. et al., 2020 |
| | | The replacement of maise (*Zea mays* L.) by cup plant (*Silphium perfoliatum* L.) as biogas substrate and its implications for the energy and material flows of a large biogas plant | Von Cossel M. et al., 2020 |
| | | Production, characterisation, and bioactivity of fish protein hydrolysates from aquaculture turbot (*Scophthalmus maximus*) wastes | Vázquez J.A., 2020 |

| | |
|---|---|
| Planning the flows of residual biomass produced by wineries for the preservation of the rural landscape | Manniello C. et al., 2020 |
| Food wastes and sewage sludge as feedstock for an urban biorefinery producing biofuels and added-value bioproducts | Battista F. et al., 2020 |
| Total replacement of dietary fish meal with black soldier fly (*Hermetia illucens*) larvae does not impair physical, chemical or volatile composition of farmed Atlantic salmon (*Salmo salar* L.) | Bruni L. et al., 2020 |
| Biorefineries: a step forward to a circular bioeconomy | Castro E., 2020 |
| Riding a Trojan horse? Future pathways of the fiber-based packaging industry in the bioeconomy | Korhonen J. et al., 2020 |
| Towards a sustainable forest-based bioeconomy in Italy: Findings from a SWOT analysis | Falcone P.M. et al., 2020 |
| Towards better life cycle assessment and circular economy: on recent studies on interrelationships among environmental sustainability, food systems and diet | Lu T. et al., 2020 |
| (Non-)Kolbe electrolysis in biomass valorisation—a discussion of potential applications | Holzhäuser F.J. et al., 2020 |
| Designing Bio-based Recyclable Polymers for Plastics | Hatti-Kaul R. et al., 2020 |
| Evaluation of the potential of alternative vegetable materials for production of paper through kraft processes | Robles J.D. et al., 2020 |
| Pilot-Scaled Fast-Pyrolysis Conversion of Eucalyptus Wood Fines into Products: Discussion Toward Possible Applications in Biofuels, Materials, and Precursors | Matos M. et al., 2020 |
| Forest Biomass Availability and Utilization Potential in Sweden: A Review | Kumar A. et al., 2020 |
| Towards a green and sustainable fruit waste valorisation model in Brazil: Optimisation of homogeniser-assisted extraction of bioactive compounds from mango waste using a response surface methodology | Zuin V.G. et al., 2020 |
| Cogrinding Wood Fibers and Tannins: Surfactant Effects on the Interactions and Properties of Functional Films for Sustainable Packaging Materials | Missio AL et al., 2020 |
| Environmental sustainability of bioenergy strategies in western Kenya to address household air pollution | Carvalho R.L. et al., 2020 |
| Valorization of linen processing by-products for the development of injection-molded green composite pieces of polylactide with improved performance | Agüero, A., 2020 |
| Environmental life cycle assessment of different biorefinery platforms valorising municipal solid waste to bioenergy, microbial protein, lactic and succinic acid | Khoshnevisan B. et al., 2020 |
| Circular Economy and Bioeconomy Interaction Development as Future for Rural Regions. Case Study of Aizkraukle Region in Latvia | Muizniece I. et al., 2019 |
| Increased utilisation of renewable resources: dilemmas for organic agriculture | Løes A.-K. et al., 2019 |
| Gasification of sewage sludge within a circular economy perspective: a Polish case study | Werle S. et al., 2019 |
| Microalgae wastewater treatment: Biological and technological approaches | Wollmann F. et al., 2019 |
| End-of-waste life: Inventory of alternative end-of-use recirculation routes of bio-based plastics in the European Union context | Briassoulis D. et al., 2019 |

| | |
|---|---|
| Advances in Food and Byproducts Processing towards a Sustainable Bioeconomy | Kopsahelis N. et al., 2019 |
| Self-sustainable azolla-biorefinery platform for valorisation of bio-based products with circular-cascading design | Hemalatha M. et al., 2019 |
| Scenedesmus obliquus microalga-based biorefinery—from brewery effluent to bioactive compounds, biofuels and biofertilisers—aiming at a circular bioeconomy | Ferreira A. et al., 2019 |
| A critical review of organic manure biorefinery models toward sustainable circular bioeconomy: Technological challenges, advancements, innovations, and future perspectives | Awasthi MK et al., 2019 |
| Bioeconomy for Sustainable Development | Aguilar A. et al., 2019 |
| A Retro-biosynthesis-Based Route to Generate Pinene-Derived Polyesters | Stamm A. et al., 2019 |
| A path transition towards a bioeconomy—The crucial role of sustainability | Gawel E. et al., 2019 |
| Risk assessments for quality-assured, source-segregated composts and anaerobic digestates for a circular bioeconomy in the UK | Longhurst P.J. et al., 2019 |
| GIS method to design and assess the transportation performance of a decentralised biorefinery supply system and comparison with a centralised system: case study in southern Quebec, Canada | Lemire P.-O. et al., 2019 |
| Circular, Green, and Bio Economy: How Do Companies in Land-Use Intensive Sectors Align with Sustainability Concepts? | D'Amato D. et al., 2019 |
| Restructuring the Conventional Sugar Beet Industry into a Novel Biorefinery: Fractionation and Bioconversion of Sugar Beet Pulp into Succinic Acid and Value-Added Coproducts | Alexandri M. et al., 2019 |
| Advances in the Use of Protein-Based Materials: Toward Sustainable Naturally Sourced Absorbent Materials | Capezza A.J. et al., 2019 |
| Food waste valorisation advocating Circular Bioeconomy—A critical review of potentialities and perspectives of spent coffee grounds biorefinery | Zabaniotou A. et al., 2019 |
| A spatial approach to bioeconomy: Quantifying the residual biomass potential in the EU-27 | Hamelin L. et al., 2019 |
| The future agricultural biogas plant in Germany: A vision | Theuerl S. et al., 2019 |
| Sequential fractionation of the lignocellulosic components in hardwood based on steam explosion and hydrotropic extraction | Olsson J. et al., 2019 |
| Assessing the forest-wood chain at local level: A multi-criteria decision analysis (MCDA) based on the circular bioeconomy principles | Pieratti E. et al., 2019 |
| Formation of theoretical and methodological assumptions in the assessment of significance of the bioeconomy in the country economy | Biekša K. et al., 2019 |
| Introduction | Klitkou, A. et al., 2019 |
| The opportunity of using chain of custody of forest-based products in the bioeconomy | Dudík R. et al., 2019 |
| Theoretical perspectives on innovation for waste valorisation in the bioeconomy | Bugge M.M. et al., 2019 |
| Plant proteins in the focus of bioeconomy | Yovchevska P., 2019 |
| Life cycle assessment: A governance tool for transition towards a circular bioeconomy? | Brekke A. et al., 2019 |

| | |
|---|---|
| Urban forests: Bioeconomy and added value | Mihailova M., 2019 |
| Future phosphorus: Advancing new 2D phosphorus allotropes and growing a sustainable bioeconomy | Jarvie H.P. et al., 2019 |
| Identifying the challenges of implementing a European bioeconomy based on forest resources: Reality demands circularity | Dimic-Misic K. et al., 2019 |
| From waste to value: Valorisation pathways for organic waste streams in circular bioeconomies | Klitkou A. et al., 2019 |
| Bio-based circular economy in European national and regional strategies | Vanhamaki S. et al., 2019 |
| A case report on inVALUABLE: Insect value chain in a circular bioeconomy | Heckmann, L.-H., 2019 |
| Circular bioeconomy in action: Collection and recycling of domestic used cooking oil through a social, reverse logistics system | Loizides M.I. et al., 2019 |
| Sustainable bioenergy policy for the period after 2020 | Šupín M. et al., 2019 |
| Selection of indicators for the assessment of national bioeconomies in the EU countries | Kakhovych E. et al., 2019 |
| Converting coffee silverskin to value-added products under a biorefinery approach | Del Pozo C. et al., 2019 |
| Conversion of crude glycerol to citric acid by yarrowia lipolytica | Giacomobono R. et al., 2019 |
| A Bio-Refinery concept for N and P recovery—A chance for biogas plant development | Szymańska M. et al., 2019 |
| Extending the design space in solvent extraction-from supercritical fluids to pressurised liquids using carbon dioxide, ethanol, ethyl lactate, and water in a wide range of proportions | Pilařová V., 2019 |
| Cross-fertilisation of ideas for a more sustainable fertiliser market: The need to incubate business concepts for harnessing organic residues and fertilisers on biotechnological conversion platforms in a circular bioeconomy | Hildebrandt J. et al., 2018 |
| Regional assessment of bioeconomy options using the anaerobic biorefinery concept | Pérez-Camacho M.N. et al., 2018 |
| Agronomic efficiency of selected phosphorus fertilisers derived from secondary raw materials for European agriculture. A meta-analysis | Huygens D. et al., 2018 |
| Residual biomass as resource—Life-cycle environmental impact of wastes in circular resource systems | Olofsson J. et al., 2018 |
| Efficient conversion of aqueous-waste-carbon compounds into electrons, hydrogen, and chemicals via separations and microbial electrocatalysis | Borole A.P. et al., 2018 |
| Lavender- and lavandin-distilled straws: An untapped feedstock with great potential for the production of high-added value compounds and fungal enzymes | Lesage-Meessen L. et al., 2018 |
| Understanding the systems that characterise the circular economy and the bioeconomy | Bezama A., 2018 |
| Bridging the gaps for a 'circular' bioeconomy: Selection criteria, bio-based value chain and stakeholder mapping | Lokesh K. et al., 2018 |
| Hydrolysis of hemicellulose and derivatives—a review of recent advances in the production of furfural | Delbecq F. et al., 2018 |
| Building an Integrative and Circular Bioeconomy | Walker L., 2018 |
| The Circular Bioeconomy—Concepts, Opportunities, and Limitations | Carus M. et al., 2018 |

| | |
|---|---|
| Bringing plant cell wall-degrading enzymes into the lignocellulosic biorefinery concept | Silva C.O.G. et al., 2018 |
| Green and Sustainable Separation of Natural Products from Agro-Industrial Waste: Challenges, Potentialities, and Perspectives on Emerging Approaches | Zuin V.G. et al., 2018 |
| EU ambition to build the world's leading bioeconomy—Uncertain times demand innovative and sustainable solutions | Bell J. et al., 2018 |
| Bio-based Industries Joint Undertaking: The catalyst for sustainable bio-based economic growth in Europe | Mengal P. et al., 2018 |
| Assessing wood use efficiency and greenhouse gas emissions of wood product cascading in the European Union | Bais-Moleman A.L. et al., 2018 |
| Review and analysis of alternatives for the valorisation of agro-industrial olive oil by-products | Berbel J. et al., 2018 |
| Bioeconomy meets the circular economy: The RESYNTEX and force projects | Leal Filho W., 2018 |
| A definition of bioeconomy through the bibliometric networks of the scientific literature | Konstantinis A. et al., 2018 |
| Towards understanding the transdisciplinary approach of the bioeconomy nexus | Muizniece I. et al., 2018 |
| New trends for mitigation of environmental impacts: A literature review | De Oliveira K.V. et al., 2018 |
| A governance framework for a sustainable bioeconomy: Insights from the case of the German wood-based bioeconomy | Gawel E. et al., 2018 |
| Chapter One: Nexus Bioenergy—Bioeconomy | Lago C. et al., 2018 |
| Modified biomass for pollution cleaning under the frames of biorefinery and sustainable circular bioeconomy | Sidiras D., 2018 |
| Recent advances in the microwave-assisted production of hydroxymethylfurfural by hydrolysis of cellulose derivatives—A review | Delbecq F. et al., 2018 |
| Bioeconomy concepts | Birner R., 2017 |
| Autotrophic biorefinery: dawn of the gaseous carbon feedstock | Butti S.K. et al., 2017 |
| Bio-based economy: Policy framework and foresight thinking | Ladu L. et al., 2017 |
| Multi-product biorefineries from lignocelluloses: A pathway to revitalisation of the sugar industry? | Farzad S. et al., 2017 |
| Cascade use indicators for selected biopolymers: Are we aiming for the right solutions in the design for recycling of bio-based polymers? | Hildebrandt J. et al., 2017 |
| Environmental and Ecological Aspects in the Overall Assessment of Bioeconomy | Székács A., 2017 |
| Analysis of the structure and values of the European Commission's Circular Economy Package | Stahel, W.R., 2017 |
| A life cycle assessment of biosolarization as a valorisation pathway for tomato pomace utilisation in California | Oldfield T.L. et al., 2017 |
| Executive summary of the report of the committee of biotechnology of the Polish academy of sciences Bioeconomy, biotechnology and new genetic engineering techniques. Modern biotechnology-based bioeconomy in a circular economy | Twardowski T., 2017 |

| | | Rural pole for competitivity: A pilot project for circular bioeconomy | Matiuti M. et al., 2017 |
|---|---|---|---|
| | | Production of Bacillus amyloliquefaciens OG and its metabolites in renewable media: valorisation for biodiesel production and p-xylene decontamination | Etchegaray A. et al., 2017 |
| | | The bioeconomy in Sicily: New green marketing strategies applied to the sustainable tourism sector | Maugeri E., 2017 |
| | | Sustainable lightweight biocomposites from toughened polypropylene and biocarbon for automotive applications | Behazin E. et al., 2017 |
| | | Food waste valorisation options: Opportunities from the bioeconomy | Imbert E., 2017 |
| | | What can be learned from practical cases of green economy? Studies from five European countries | Pitkänen K. et al. 2016 |
| | | Making the Bioeconomy Circular: The Bio-based Industries' Next Goal? | Sheridan K., 2016 |
| | | Waste biorefinery models towards sustainable circular bioeconomy: Critical review and future perspectives | Venkata Mohan S. et al., 2016 |
| | | Recovery of Resources From Biowaste for Pollution Prevention | Prasad M.N.V., 2016 |
| | | Regulatory policies and trends | Kurppa, S., 2015 |
| | | The political economy of fostering a wood-based bioeconomy in Germany | Pannicke N. et al., 2015 |
| | | Research and innovation in agriculture: Beyond productivity? | Viaggi D., 2015 |
| ScienceDirect | Circular bioeconomy business model biomass | Transition in the Finnish forest-based sector: Company perspectives on the bioeconomy, circular economy and sustainability | Näyhä A., 2019 |
| ScienceDirect | Bioeconomy business model biomass | Oil palm biomass biorefinery for future bioeconomy in Malaysia | Mohd Yusof SJH et al., 2017 |
| | | High-value low-volume bioproducts coupled to bioenergies with potential to enhance business development of sustainable biorefineries | Budzianowski, W.M., 2017 |
| ScienceDirect | Circular bioeconomy business model | Towards sustainability? Forest-based circular bioeconomy business models in Finnish SMEs | D'Amato D. et al., 2020 |
| | | Finnish forest-based companies in transition to the circular bioeconomy—drivers, organisational resources and innovations | Näyhä A., 2020 |
| | | Squaring the circle: Refining the competitiveness logic for the circular bioeconomy | DeBoer J. et al., 2020 |
| | | The circular economy and the bio-based sector—Perspectives of European and German stakeholders | Leipold S. et al., 2018 |
| | | Waste-derived bioeconomy in India: A perspective | Venkata Mohan S. et al., 2018 |
| ScienceDirect | Bieconomy business model | A transition to an innovative and inclusive bioeconomy in Aragon, Spain | Sanz-Hernández A., 2019 |
| | | Application of multi criteria analysis methods for a participatory assessment of non-wood forest products in two European case studies | Huber P. et al., 2019,2018 |
| | | Digital solutions transform the forest-based bioeconomy into a digital platform industry—A suggestion for a disruptive business model in the digital economy | Watanabe C. et al. |
| | | From opportunities to action—An integrated model of small actors' engagement in bioenergy business | Kokkonen K. et al., 2018 |
| | | Forest biorefinery: Potential of poplar phytochemicals as value-added co-products | Devappa R.K. et al., 2015 |

| | | Resource recovery from wastewaters using microalgae-based approaches: A circular bioeconomy perspective | Nagarajan D. et al., 2020 |
|---|---|---|---|
| | | Sustainable food waste management towards circular bioeconomy: Policy review, limitations and opportunities | Tiffany M. W. Mak et al., 2020 |
| | | Do forest biorefineries fit with working principles of a circular bioeconomy? A case of Finnish and Swedish initiatives | Temmes A. et al., 2020 |
| | | Bioconversion of waste (water)/residues to bioplastics—A circular bioeconomy approach | Yadav B. et al., 2020 |
| | | Sustainable production of bio-based chemicals and polymers via integrated biomass refining and bioprocessing in a circular bioeconomy context | Ioannidou S.M. et al., 2020 |
| | | Biorefineries in circular bioeconomy: A comprehensive review | Ubando A.T. et al., 2020 |
| | | Biorefinery of spent coffee grounds waste: Viable pathway towards circular bioeconomy | Rajesh Banu J. et al., 2020 |
| | | Food waste and social acceptance of a circular bioeconomy: the role of stakeholders | Morone P. et al., 2020 |
| | | The role of the policy mix in the transition toward a circular forest bioeconomy | Ladu L. et al., 2020 |
| | | Microalgae based biorefinery promoting circular bioeconomy-techno economic and life-cycle analysis | Rajesh Banu J., 2020 |
| **ScienceDirect** | **Circular bioeconomy** | Valorisation of waste eggshell-derived bioflocculant for harvesting T. obliquus: Process optimisation, kinetic studies and recyclability of the spent medium for circular bioeconomy | Roy M. et al., 2020 |
| | | A perspective on decarbonising whiskey using renewable gaseous biofuel in a circular bioeconomy process | Kang X. et al., 2020 |
| | | Strategic decisions on knowledge development and diffusion at pilot and demonstration projects: An empirical mapping of actors, projects and strategies in the case of circular forest bioeconomy | Hedeler B. et al., 2020 |
| | | Towards a more sustainable circular bioeconomy. Innovative approaches to rice residue valorisation: The RiceRes case study | Overturf E., Ravasio N. et al., 2020 |
| | | Forest-based circular bioeconomy: matching sustainability challenges and novel business opportunities? | Toppinen A. et al., 2020 |
| | | Bio-combustion of petroleum coke: The process integration with photobioreactors. Part II—Sustainability metrics and bioeconomy | Severo I.A., 2020 |
| | | Algal biorefinery models with self-sustainable closed loop approach: Trends and prospective for blue-bioeconomy | Venkata Mohan S. et al., 2020 |
| | | Towards transparent valorisation of food surplus, waste and loss: Clarifying definitions, food waste hierarchy, and role in the circular economy | Teigiserova D.A., 2020 |
| | | A review of LCA assessments of forest-based bioeconomy products and processes under an ecosystem services perspective | D'Amato D. et al., 2020 |
| | | Co-evolutionary coupling leads the way to a novel concept of R&D—Lessons from digitalised bioeconomy | Naveed N. et al., 2020 |
| | | Friends or foes? A compatibility assessment of bioeconomy-related Sustainable Development Goals for European policy coherence | Ronzon T. et al., 2020 |

| | |
|---|---|
| The significance of biomass in a circular economy | Sherwood J., 2020 |
| A perspective on novel cascading algal biomethane biorefinery systems | Bose A. et al., 2020 |
| Innovative integrated approach of biofuel production from agricultural wastes by anaerobic digestion and black soldier fly larvae | Elsayed M. et al., 2020 |
| Life Cycle Assessment of specific organic waste-based bioeconomy approaches | Smetana S., 2020 |
| Co-digestion of by-products and agricultural residues: A bioeconomy perspective for a Mediterranean feedstock mixture | Valenti F. et al., 2020 |
| Recent advances on the sustainable approaches for conversion and reutilization of food wastes to valuable bioproducts | Hui Suan Ng et al., 2020 |
| Biomolecules from municipal and food industry wastes: An overview | Lee J.K. et al., 2020 |
| Green processes in Foodomics. Supercritical Fluid Extraction of Bioactives | Mazzutti S. et al., 2020 |
| Sustainable valorisation of sugar industry waste: Status, opportunities, and challenges | Meghana M. et al., 2020 |
| Recent developments in microalgal conversion of organics-enriched waste streams | Solovchenko A. et al., 2020 |
| Biorefineries for the valorisation of food processing waste | Moreno D. A. et al., 2020 |
| Chapter 8: Chemical and energy potential of sugarcane | Rabelo S. C. et al., 2020 |
| Cellulose-Derived Hydrothermally Carbonized Materials and their Emerging Applications | Adolfsson C. H. et al., 2020 |
| Transforming the bio-based sector towards a circular economy—What can we learn from wood cascading? | Jarre M. et al., 2020 |
| Recovery of high value-added compounds from pineapple, melon, watermelon and pumpkin processing by-products: An overview | Rico X. et al., 2020 |
| Chapter 3: Triple bottom line, sustainability and sustainability assessment, an overview | Sala S., 2020 |
| 18: Food industry waste biorefineries | Kumar P. S. et al., 2020 |
| Chapter 8: Life cycle assessment of waste-to-bioenergy processes: a review | Ghosh P. et al., 2020 |
| Influence of green solvent on levulinic acid production from lignocellulosic paper waste | Dutta S. et al., 2020 |
| Enhanced nitrogen removal of low carbon wastewater in denitrification bioreactors by utilising industrial waste toward circular economy | Kiani S. et al., 2020 |
| On the Circular Bioeconomy and Decoupling: Implications for Sustainable Growth | Giampietro M., 2019 |
| Can circular bioeconomy be fueled by waste biorefineries—A closer look | Mohan S. V. et al., 2019 |
| Green Bioplastics as Part of a Circular Bioeconomy | Karan H. et al., 2019 |
| Microalgal Aquafeeds As Part of a Circular Bioeconomy | Yarnold J. et al., 2019 |
| Whey and molasses as inexpensive raw materials for parallel production of biohydrogen and polyesters via a two-stage bioprocess: New routes towards a circular bioeconomy | Carlozzi P. et al., 2019 |
| Digitalised bioeconomy: Planned obsolescence-driven circular economy enabled by Co-Evolutionary coupling | Watanabe C. et al., 2019 |

| | | |
|---|---|---|
| | Review of high-value food waste and food residues biorefineries with focus on unavoidable wastes from processing | Teigiserova D.A. et al., 2019 |
| | Thinking green, circular or bio: Eliciting researchers' perspectives on a sustainable economy with Q method | D'Amato D. et al., 2019 |
| | Characteristics of bioeconomy systems and sustainability issues at the territorial scale. A review | Wohlfahrt J. et al., 2019 |
| | Multiproduct biorefinery from Arthrospira spp. towards zero waste: Current status and future trends | Mitra M. et al., 2019 |
| | Biodiesel facilities: What can we address to make biorefineries commercially competitive? | Severe I.A. et al., 2019 |
| | Chapter One: Nexus Bioenergy–Bioeconomy | Lago C. et al., 2019 |
| | Bioprocess development for the production of novel oleogels from soybean and microbial oils | Papadaki A. et al., 2019 |
| | Novel insights and innovations in biotechnology towards improved quality of life | Barciszewski J. et al., 2019 |
| | Efficient resource valorisation by co-digestion of food and vegetable waste using three stage integrated bioprocess | Chakraborty D. et al., 2019 |
| | Chapter 3—Systems Analysis Frameworks for Biorefineries | Murthy G. S. et al., 2019 |
| | Microalgal bioenergy production under zero-waste biorefinery approach: Recent advances and future perspectives | Mishra S., 2019 |
| | Advances in lignin valorisation towards bio-based chemicals and fuels: Lignin biorefinery | Cao Y. et al., 2019 |
| | Chemical composition and biological activities of Juçara (Euterpe edulis Martius) fruit by-products, a promising underexploited source of high-added value compounds | Garcia J.A.A. et al., 2019 |
| | Evolution and perspectives of the bioenergy applications in Spain | Paredes-Sánchez J. P. et al., 2019 |
| | Chapter 10: Vermicomposting of Waste: A Zero-Waste Approach for Waste Management | Sharma K. et al., 2019 |
| | Chapter 19: Utilisation and Management of Horticultural Waste | Lobo M.G. et al., 2019 |
| | Redesigning a bioenergy sector in EU in the transition to circular waste-based bioeconomy: A multidisciplinary review | Zabaniotou A., 2018 |
| | Food waste biorefinery: Sustainable strategy for circular bioeconomy | Dahiya S. et al., 2018 |
| | Biowaste Valorisation in a Future Circular Bioeconomy | Vea E.B. et al., 2018 |
| | Taking a reflexive TRL3–4 approach to sustainable use of sunflower meal for the transition from a mono-process pathway to a cascade biorefinery in the context of Circular Bioeconomy | Zabaniotou A. et al., 2018 |
| | Multi-scale system modelling under circular bioeconomy | Guo M., 2018 |
| | Sidestreams from bioenergy and biorefinery complexes as a resource for circular bioeconomy | Konwar L.J. et al., 2018 |
| | Combining biotechnology with circular bioeconomy: From poultry, swine, cattle, brewery, dairy and urban wastewaters to biohydrogen | Ferreira A. et al., 2018 |
| | An efficient agro-industrial complex in Almería (Spain): Towards an integrated and sustainable bioeconomy model | Egea F. J. et al., 2018 |
| | Spanish strategy on bioeconomy: Towards a knowledge based sustainable innovation | Lainez M. et al., 2018 |

| | Consensus, caveats and conditions: International learnings for bioeconomy development | Devaney L. et al., 2018 |
|---|---|---|
| | Destination bioeconomy—The path towards a smarter, more sustainable future | Dupont-Inglis J. et al., 2018 |
| | Sustainable bioeconomy transitions: Targeting value capture by integrating pyrolysis in a winery waste biorefinery | Zabaniotou A. et al., 2018 |
| | Forest sector circular economy development in Finland: A regional study on a sustainability-driven competitive advantage and an assessment of the potential for cascading recovered solid wood | Husgafvel R. et al., 2018 |
| | Cascading Norwegian co-streams for bioeconomic transition | Egelyng H. et al., 2018 |
| | Separation of value-added chemical groups from bio-oil of olive mill waste | Del Pozo C. et al., 2018 |
| | Role of bioenergy, biorefinery and bioeconomy in sustainable development: Strategic pathways for Malaysia | Sadhukhan J. et al., 2018 |
| | The role of biogas solutions in sustainable biorefineries | Hagman L. et al., 2018 |
| | Initial indicator analysis of bioethylen production pathways | Kuznecova I. et al., 2018 |
| | Nutrient management via struvite precipitation and recovery from various agroindustrial wastewaters: Process feasibility and struvite quality | Taddeo R. et al., 2018 |
| | Residual biomass as resource—Life-cycle environmental impact of wastes in circular resource systems | Olofsson J. et al., 2018 |
| | Valorisation of polyhydroxyalkanoates production process by co-synthesis of value-added products | Kumar P. et al., 2018 |
| | Life cycle environmental impacts of substituting food wastes for traditional anaerobic digestion feedstocks | Pérez-Camacho M.N. et al., 2018 |
| | Techno-economic and profitability analysis of food waste biorefineries at European level | Cristóbal J. et al., 2018 |
| | How is the term 'ecotechnology' used in the research literature? A systematic review with thematic synthesis | Haddaway N.R. et al., 2018 |
| | A roadmap towards a circular and sustainable bioeconomy through waste valorisation | Maina S. et al., 2017 |
| | Green, circular, bio economy: A comparative analysis of sustainability avenues | D'Amato D. et al., 2017 |
| | Opportunities for bioenergy in the Baltic Sea Region | Silveira S. et al., 2017 |
| | Olive mill solid waste biorefinery: High-temperature thermal pre-treatment for phenol recovery and biomethanization | Serrano A. et al., 2017 |
| | The suitability of banana leaf residue as raw material for the production of high lignin content micro/nano fibers: From residue to value-added products | Tarrés Q. et al., 2017 |
| | A life cycle assessment of biosolarization as a valorisation pathway for tomato pomace utilisation in California | Oldfield T.L. et al., 2017 |
| | Life cycle assessment of wood-plastic composites: Analysing alternative materials and identifying an environmental sound end-of-life option | Sommerhuber P.F. et al., 2017 |
| | A Circular Bioeconomy with Biobased Products from CO2 Sequestration | Venkata Mohan S. et al., 2016 |

| | | Waste Biorefinery: A New Paradigm for a Sustainable Bioelectro Economy | Venkata Mohan S. et al., 2016 |
|---|---|---|---|
| | | An environmental analysis of options for utilising wasted food and food residue | Oldfield T.L. et al., 2016 |
| | | Life cycle assessment of macroalgal biorefinery for the production of ethanol, proteins and fertilisers—A step towards a regenerative bioeconomy | Seghetta M. et al., 2016 |
| | | Strategy and design of Innovation Policy Road Mapping for a waste biorefinery | Rama Mohan S., 2016 |
| | | Sustainability of biofuels and renewable chemicals production from biomass | Kircher M., 2015 |
| | | Assessing support of pilot production in multi-KETs activities | Butter M. et al., 2015 |
| | | Innovation Ecosystems in the EU: Policy Evolution and Horizon Europe Proposal Case Study (the Actors' Perspective) | Fernández S. G. et al., 2019 |
| | | Optimisation models for financing innovations in green energy technologies | Tan R.R. et al., 2019 |
| | | Circular Business Models for the Bio-Economy: A Review and New Directions for Future Research | Wiebke R. et al., 2019 |
| **Other** | **Bioeconomy business model** | Bioeconomy mapping report An overview of the bioeconomy | Bos H. et al., 2018 |
| | | Literature Review: Investment Readiness Level of Small and Medium Sized Companies | Fellnhofer K., 2016 |
| | | Resources, collaborators, and neighbors: The three-pronged challenge in the implementation of bioeconomy regions | Bezama A. et al., 2019 |
| | | Technological innovation systems for biorefineries: a review of the literature | Bauer F. et al., 2017 |
| | | New innovative ecosystems in France to develop the Bioeconomy | Stadler T. et al., 2016 |
| | | Overcoming "The Valley of Death" | Mcintyre R.A., 2014 |
| | | The Making of BIOECONOMY TRANSFORMATION | Kruus K. et al., 2017 |
| **Other** | **Circular bioeconomy** | The German R&D Program for $CO_2$ Utilisation—Innovations for a Green Economy | Lothar Mennicken et al., 2016 |
| | | Potential of biomass sidestreams for a sustainable biobased economy | Cabeza C. et al., 2019 |
| | | Barriers and incentives for the use of lignin-based resins: Results of a comparative importance performance analysis | Lettner M. et al., 2020 |
| | | Sustainability Indicators for Biobased Product Manufacturing: A systematic review | Kooduvalli K. et al., 2019 |

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
