# Peer review of "Circular Bioeconomy Business Models to Overcome the Valley of Death. A Systematic Statistical Analysis of Studies and Projects in Emerging Bio-Based Technologies and Trends Linked to the SME Instrument Support"

_sustainability, doi:10.3390/su13041899_

Round 1

Reviewer 1 Report

The manuscript is well written. I have only some suggests for the authors.

The title seems too long. It is explanatory but not very attractive for searches. Figures can be improved (font color black, font size bigger, while colours choices ar good).

All those digits are needed (including decimals) in figue 6. It could be improved with scientific notation.

Author Response

Dear Reviewer,

Reviewer 2 Report

The topic of the manuscript is relevant and meet the scope of a journal. Although the paper contains some drawbacks that must be addressed before publication.

1) The introduction should more precisely stress the necessity of the research, its aim originality and indicate the prevailing theoretical streams this paper is aimed to enrich.

2) The more broad theoretical background should be provided. In case of literature reviews (as it is the case) it is expected to see a brief of literature indicating the main research directions in an investigated area and show some scientific vacuum which is being aimed to fulfill with current manuscript.

3) The discussion section contain a lot of information not mentioned/investigated previously in the paper. It is not a welcomed approach.

4) The conclusion part should be enhanced and deepened. Future research directions should be provided.

Author Response

Dear Reviewer, 

Reviewer 3 Report

Dear Authors, thank you for your very interesting work! Well done!

Author Response

Dear reviewer,

Thank you for your comments concerning our manuscript entitled “Circular bioeconomy business models to overcome the Valley of death. A systematic statistical analysis of studies and research projects in emerging biobased technologies and trends linked to the SME Instrument financial support” (ID: sustainability-1084430).

Thank you for your interest in research work and the excellent evaluation provided.

We would like to inform you that a reviewer has requested further revisions. The revised manuscript is at your disposal in case you want to see these comments.

Round 2

Reviewer 2 Report

The authors made a god job polishing the paper. All my comments were addressed properly. I recommend this manuscript to be published.